# Learning the Complexity of Weakly Noisy Quantum States

**Yusen Wu**[1,2]**, Bujiao Wu**[*3,4,5,6]**, Yanqi Song**[7]**, Xiao Yuan**[6] **& Jingbo B. Wang**[†2]

1 School of Artificial Intelligence, Beijing Normal University, Beijing, 100875, China
2 Department of Physics, The University of Western Australia, Perth, WA 6009, Australia
3 Shenzhen Institute for Quantum Science and Engineering,
Southern University of Science and Technology, Shenzhen 518055, China
4 International Quantum Academy, Shenzhen 518048, China
5 Dahlem Center for Complex Quantum Systems, Freie Universität Berlin, 14195 Berlin, Germany
6 Center on Frontiers of Computing Studies, Peking University, Beijing 100871, China
7 China Academy of Information and Communications Technology, Beijing, 100191, China
*bujiaowu@gmail.com, †jingbo.wang@uwa.edu.au

## Abstract

Quantifying the complexity of quantum states is a longstanding key problem in various subfields of science, ranging from quantum computing to the black-hole theory. The lower bound on quantum pure state complexity has been shown to grow linearly with system size (Haferkamp et al., 2022). However, extending this result to noisy circuit environments, which better reflect real quantum devices, remains an open challenge. In this paper, we explore the complexity of weakly noisy quantum states via the quantum learning method. We present an efficient learning algorithm, that leverages the classical shadow representation of target quantum states, to predict the circuit complexity of weakly noisy quantum states. Our algorithm is proved to be optimal in terms of sample complexity accompanied with polynomial classical processing time. Our result builds a bridge between the learning algorithm and quantum state complexity, meanwhile highlighting the power of learning algorithm in characterizing intrinsic properties of quantum states.

## 1 Introduction

The concept of quantum complexity has deep connections to high-energy physics, quantum many-body systems, and black-hole physics (Bouland et al., 2019b; Brown et al., 2016a;b; Stanford & Susskind, 2014; Susskind, 2016). A problem is deemed "easy" if it is soluble with a (quantum) circuit whose size grows polynomially with the size of the inputs, while the problem is deemed "hard" if the size of the (quantum) circuit scales exponentially. In the context of quantum computation, the complexity of an $n$-qubit quantum state corresponds to the minimum number of gates required to prepare it from the initial state $|0^n\rangle$ (Haferkamp et al., 2022). Brown and Susskind's conjecture suggests that the complexity of quantum states generated by random quantum circuits grows linearly before it saturates after reaching an exponential size (Brown & Susskind, 2018; Susskind, 2018), which is supported by the complexity geometry theory proposed by Nielsen et al. (2006). Recent works theoretically proved this conjecture by connecting quantum state complexity to unitary $t$-designs (Brandão et al., 2021; Jian et al., 2022) and the dimension of semi-algebraic sets (Haferkamp et al., 2022), demonstrating a rigorous computational ability separation between shallow and deep quantum circuits.

However, in the practical world, the quantum system may interact with the surrounding environment, which inevitably introduces a noise signal, making the pure state noisy. For example, in the current noisy-intermediate-scale-quantum device (Boixo et al., 2018; Arute et al., 2019; Zhong et al., 2020; Wu et al., 2021), a certain level (constant) of noise exists in each quantum gate, resulting in noisy states prepared by $\Omega(\log n)$-depth noisy circuits is classically simulable in both mean value computation and random circuit sampling problems (Stilck França & Garcia-Patron, 2021; Aharonov et al., 2022). These facts exhibit a trend that is completely different from the case of pure quantum

states: increasing the depth of the noisy circuit reduces the quantum state complexity. However, it is still unclear how to extend previous arts (Brandão et al., 2021; Jian et al., 2022; Haferkamp et al., 2022) to the noisy environment, and how to characterize the noisy quantum state complexity is still an open problem.

On the other hand, learning algorithms have recently been considered a powerful means to study and understand many-particle quantum systems and the associated quantum processes (Carleo & Troyer, 2017; Carrasquilla & Melko, 2017; Glasser et al., 2018; Torlai et al., 2018; Moreno et al., 2020; Torlai & Melko, 2016; Schindler et al., 2017; Greplova et al., 2020; Wetzel, 2017; Huang et al., 2022; Wu et al., 2023b; Du et al., 2022). Specifically, Huang et al. (2022) explored the power of the learning algorithm in classifying quantum phases of matter by learning through the classical shadow (Huang et al., 2020), an efficient classical representation of the quantum state. The strong connection between quantum state complexity and quantum phase transition phenomena Huang et al. (2015) motivates us to explore the complexity of noisy quantum states using a learning algorithm.

Here, we focus primarily on the complexity of *weakly noisy quantum states* to avoid the anti-concentration property (Deshpande et al., 2022), an undesirable distribution in quantum simulation and quantum approximate optimization algorithms. The $n$-qubit weakly noisy quantum states are generated by circuits with a depth of $\tilde{R} \leq O(\text{poly}\log n)$ and a noise strength of $\mathcal{O}(1/n)$. [1] We focus on an essential and natural problem: *Given* $\text{poly}(n)$ *copies of* $n$-*qubit unknown weakly noisy quantum state, how to predict its complexity?*

To answer this question, we develop a learning approach to solve this problem, which is illustrated in Fig. 1. Following the strong quantum state complexity proposed by Brandão et al. (2021), the learning algorithm pertains to determine *whether an unknown weakly noisy state* $\rho$ [2] *can be* $\epsilon$-*distinguished from the maximally mixed state by a measurement operator induced by a specific quantum circuit architecture (QCA) (shown in Fig. 1 (b))*. The size of the measurement operator provides a state complexity upper bound (Brandão et al., 2021). More specifically, the problem asks whether there exists a measurement operator $M$ induced by a specific QCA, such that

$$\max_{\substack{M=U|0^n\rangle\langle 0^n|U^\dagger \\ U\in\mathcal{U}_\mathcal{A}(R)}} |\text{Tr}\left(M(\rho - I_n/2^n)\right)| \geq \Omega(1-\epsilon), \tag{1}$$

where $\mathcal{A}$ denotes a QCA architecture and $\mathcal{U}_\mathcal{A}(R)$ contains all $R$-depth QCA circuits induced by $\mathcal{A}$. The quantum state complexity can be bounded by $\mathcal{O}(nR)$, provided that Eq. 1 is satisfied. In general, the circuit set $\mathcal{U}_\mathcal{A}$ contains an exponential number of candidate circuits $U$, making direct enumeration impractical. We address this challenge by revealing a specific property of QCA circuits, as outlined in Theorem 1. This property allows the behavior of an observable $M$, generated by any circuit in $\mathcal{U}_\mathcal{A}$, to be approximated by a linear combination of $\text{poly}(n)$ random QCA circuits from $\mathcal{U}_\mathcal{A}$. Leveraging this, we reformulate the complexity prediction problem into the optimization of a linear function over a compact set, which is both sample-efficient and computationally efficient, as demonstrated in Theorem 2 and illustrated in Fig. 1 (c). We finally show the sample complexity of our learning algorithm is *optimal* with respect to the noisy circuit depth $\tilde{R}$, as demonstrated in Theorem 3.

We note that learning the quantum state complexity has broad applications Firstly, the predicted complexity enables the classification of the target unknown state $\rho$ into one of the following scenarios: (i) $\rho$ can be approximated by an estimator generated by constant-depth noiseless circuits, which is amenable to classical simulation (Napp et al., 2022; Bravyi et al., 2021), (ii) $\rho$ can be approximated using noiseless circuits with sub-logarithmic depth, a task that may present classical computational challenges, as discussed in Ref. (Deshpande et al., 2022), and (iii) $\rho$ cannot be approximated by any linear combination of circuits constrained to a specific architecture with a depth of at most $\log n$. Besides benchmarking the NISQ computational power (whether its output can be classically simulated), the proposed quantum algorithm and predicted weakly noisy state complexity may have

---

[1] The noise strength $\mathcal{O}(1/n)$ follows the settings used in quantum error mitigation algorithms (Temme et al., 2017; Endo et al., 2018).

[2] Here, the analyzed weakly noisy quantum states may represent ground states of many-body systems, the output state of a noisy NISQ algorithm, or the boundary state of a black hole, which all can be characterized through shadow tomography (Huang et al., 2020), as depicted in Fig. 1 (a).

practical applications in various fields, including understanding the black hole theory and the quantum phase transition. For example, in the context of the Anti-de-Sitter-space/Conformal Field Theory (AdS/CFT) correspondence, the "complexity equals volume" conjecture (Stanford & Susskind, 2014) suggests that the boundary state of the correspondence has a complexity proportional to the volume behind the event horizon of a black hole in the bulk geometry. However, when measuring the boundary state, the interaction with the surrounding environment inevitably introduces a noise signal, making the pure state noisy. Furthermore, in quantum many-body systems, quantum phase transitions occur when the external parameters varies (Sachdev, 1999), and the ability to correctly predict the quantum phase transition boundary can help us understand many strong-correlated systems (Zheng et al., 2017). It is known that quantum topological phases can be distinguished by their ground state complexity (Huang et al., 2015), and the shadow tomography (Huang et al., 2020; 2022) method utilized quantum channels to provide a noisy state approximation to the ground state. Predicting complexity of such noisy ground state approximations may recognize the topological phases of matter.

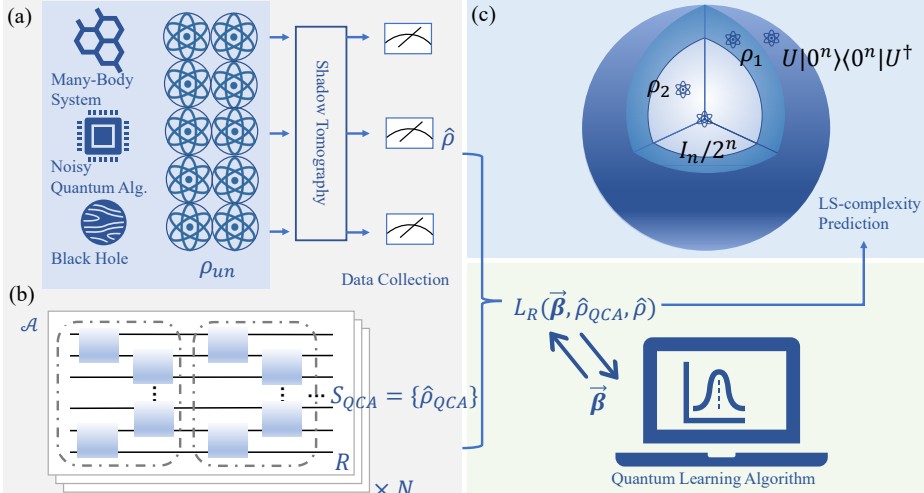

Figure 1: (a) An efficient quantum-to-classical representation conversion method that leverages the classical shadow of a noisy state (Huang et al., 2020). By measuring $\text{poly}(n)$ copies of the state, it can construct a classical representation $\hat{\rho}$ that enables the prediction of the state's properties with a rigorous performance guarantee. Here, $\rho_{\text{un}}$ represents a noisy state generated by a $\tilde{R}$-depth noisy quantum circuit model which is defined as Def. 2, and $\hat{\rho}$ represents the classical shadow of $\rho_{\text{un}}$. (b) A QCA model with $R$ blocks, where each block corresponds to a causal slice, and the overall architecture follows the design of $\mathcal{A}$. (c) Visualization of the complexity prediction process via a quantum learning algorithm (Alg. 1), where $\mathcal{L}_R(\vec{\beta}, \hat{\rho}_{\text{QCA}}, \hat{\rho})$ represents a metric in measuring the distance between $\hat{\rho}$ and $\hat{\rho}_{\text{QCA}}$ states, which is defined in Eq. 11. The blue Bloch sphere illustrates the relationship between a $\tilde{R}$-block weakly noisy quantum state $\hat{\rho}$ and its nearest pure state. All pure states reside on the surface of the $n$-qubit Bloch sphere, with the maximum mixed state $I_n/2^n$ located at the center of the sphere. In the regime where $\tilde{R} < \mathcal{O}(\log n)$ and for a small noise strength $p < 1/n$, the weakly noisy state $\hat{\rho} = \rho_1$ is located near the surface of the sphere, while $\hat{\rho} = \rho_2$ locates near the maximum mixed state.

## 2 THEORETICAL BACKGROUND

To clearly demonstrate the motivation and contribution of this work, we review the related theoretical backgrounds in terms of quantum state complexity, noisy quantum states, and the architecture of quantum circuits.

Here, we consider the quantum state complexity of an $n$-qubit quantum pure state $|\psi\rangle$. The complexity of a quantum state is the minimal circuit size required to implement a measurement operator that

suffices to distinguish $|\psi\rangle\langle\psi|$ from the maximally mixed state $I_n/2^n$. Since any pure state $|\psi\rangle\langle\psi|$ satisfies $\frac{1}{2}\||\psi\rangle\langle\psi| - I_n/2^n\|_1 = 1 - 2^{-n}$, which is achieved by the optimal measurement strategy $M = |\psi\rangle\langle\psi|$, such trace distance can be used to quantify the quantum state complexity. Let $\mathbb{H}_{2^n}$ denote the space of $2^n \times 2^n$ Hermitian matrices, and for fixed $c \in \mathbb{N}$, we consider a class of measurement operators $M_c(2^n) \subset \mathbb{H}_{2^n}$ that can be constructed by at most $c$ 2-local gates. The maximal bias achievable for quantum states with such a restricted set of measurements of the solution is defined as:

$$\xi_{QS}(c, |\psi\rangle) = \max_{M \in M_c(2^n)} |\mathrm{Tr}\left(M(|\psi\rangle\langle\psi| - I_n/2^n)\right)|, \tag{2}$$

where $|\psi\rangle = V|0^n\rangle$ for some $V \in \mathrm{SU}(2^n)$. Noting that the above metric $\xi_{QS}(c, |\psi\rangle)$ degenerates to $1 - 2^{-n}$ when $c \to \infty$. For example, if the quantum state $|\psi\rangle$ can be easily prepared by a quantum computer (such as computational basis states), $\xi_{QS}(c, |\psi\rangle)$ converges to $1 - 2^{-n}$ rapidly as $c$ increases. In contrast, for a general quantum state $|\psi\rangle$, $\xi_{QS}$ requires an exponentially large $c$ to approach $1 - 2^{-n}$. Using this property, the quantum state complexity can be defined as follows:

**Definition 1 (Approximate State Complexity (Brandão et al., 2021))** *Given an integer $c$ and $\epsilon \in (0, 1)$, we say a pure quantum state $|\psi\rangle$ has $\epsilon$-strong state complexity at most $c$ if and only if*

$$\xi_{QS}(c, |\psi\rangle) \geq 1 - \frac{1}{2^n} - \epsilon, \tag{3}$$

*which is denoted as $C_\epsilon(|\psi\rangle) \leq c$.*

Due to imperfections in quantum hardware devices, two causal slices [3] are separated by a quantum channel $\mathcal{E}$. In this paper, we consider $\mathcal{E}$ to represent a general noise channel which is both gate-independent and time-invariant, such as the local-depolarizing channel, the global-depolarizing channel, the bit-flip channel and other common noise models.

**Definition 2 (Weakly Noisy Quantum State)** *We assume that the noise in the quantum device is modeled by a gate-independent Pauli noise channel $\mathcal{E}$ with error rate $p$. Let $\mathcal{U}$ be a causal slice, and let $\mathcal{E} \circ \mathcal{U}$ be the representation of a noisy gate. We define the $\tilde{R}$-depth noisy quantum state with noise strength $p$ as*

$$\rho_{p,\tilde{R}} = \mathcal{E} \circ \mathcal{U}_{\tilde{R}} \circ \mathcal{E} \circ \mathcal{U}_{\tilde{R}-1} \circ \cdots \circ \mathcal{E} \circ \mathcal{U}_1(|0^n\rangle\langle 0^n|). \tag{4}$$

*We use the term "weakly noisy quantum states" to refer to noisy states $\rho_{p,\tilde{R}}$ with $\tilde{R} \leq \mathcal{O}(\mathrm{poly}\log n)$ and small error rate $p$ such that $p < \mathcal{O}(n^{-1})$.*

## 3 PROBLEM STATEMENT

### 3.1 WEAKLY NOISY STATE COMPLEXITY

In this paper, we assume multiple copies of a weakly noisy quantum state $\rho$ are provided, and we utilize a learning approach to predict its quantum state complexity, accessing only one copy at a time. In the learning phase, we don't have any information on $\rho$, as a result, it is generally difficult to exclude shortcuts that could improve the efficiency of a computation. As a result, deriving quantum complexity measures for weakly noisy states may be challenging without additional assumptions. Here, we supplement limitations to the operator architecture, which leads to the limited-structured quantum state complexity.

**Definition 3 (Limited-Structured (LS) Complexity of Weakly Noisy State)** *Given an integer $c$ and $\epsilon \in (0, 1)$, we say a weakly noisy state $\rho$ has $\epsilon$-LS complexity at most $c$ if and only if*

$$\max_{\substack{M_c = U|0^n\rangle\langle 0^n|U^\dagger \\ U \in \mathcal{U}_\mathcal{A}([c/L])}} |\mathrm{Tr}\left(M_c(\rho - I_n/2^n)\right)| \geq 1 - \frac{1}{2^n} - \epsilon \tag{5}$$

*which is denoted as $C_\epsilon^{\mathrm{lim},\mathcal{A}}(\rho) \leq c$. The notation $L$ represents the number of gates in each layer of $U \in \mathcal{U}_\mathcal{A}$[4], and $\epsilon$ is termed as the LS error.*

---

[3]Details refer to Appendix A.

[4]We leave rigorous definitions to Def. 6.

The measurement operator $M_c$ (in Def 3) is limited by the architecture $\mathcal{U}_\mathcal{A}$. It is interesting to note that the LS complexity provides an upper bound for $C_\epsilon(\rho_{\text{un}})$, that is,

$$C_\epsilon(\rho) \leq C_\epsilon^{\text{lim},\mathcal{A}}(\rho) \leq c.$$

Now, we provide an insight on why Def. 1 can be modified to Def. 3 which characterized the weakly noisy quantum state complexity.

**Fact 1** *Suppose we are given a general noise channel $\mathcal{E}(\cdot) = \sum_{l=1}^K K_l(\cdot)K_l^\dagger$ and a $\tilde{R}$-depth noisy state $\rho_{\tilde{R}} = \bigcirc_{r=1}^{\tilde{R}} \mathcal{E} \circ \mathcal{U}_r(|0^n\rangle\langle 0^n|)$, where the $K_l$ represent $n$-qubit Kraus operators and the $\mathcal{U}_r$ are drawn independently from a unitary 2-design set $\mathbb{U}$. If the following relationship holds:*

$$\tilde{R} \leq \frac{\log\left(\frac{F-1}{d(d+1)}(\eta - 1/d)^{-1}\right)}{\log(d^2 - 1) - \log(F - 1)}, \tag{6}$$

*where $d = 2^n$ and $F = \sum_{l=1}^K |\text{Tr}(K_l)|^2$, then for each noisy state $\rho_{\tilde{R}}$, their corresponding $\tilde{R}$-depth pure state $|\Psi_{\mathcal{U}_1,\ldots,\mathcal{U}_{\tilde{R}}}\rangle = \bigcirc_{r=1}^{\tilde{R}} \mathcal{U}_r(|0^n\rangle\langle 0^n|)$ satisfies*

$$\mathbb{E}_{\mathcal{U}_1,\ldots,\mathcal{U}_{\tilde{R}}\sim\mathbb{U}}\left[\langle\Psi_{\mathcal{U}_1,\ldots,\mathcal{U}_{\tilde{R}}}|\left(\bigcirc_{r=1}^{\tilde{R}}\mathcal{E}\circ\mathcal{U}_r(|0^n\rangle\langle 0^n|)\right)|\Psi_{\mathcal{U}_1,\ldots,\mathcal{U}_{\tilde{R}}}\rangle\right] \geq \eta. \tag{7}$$

*Specifically, $\tilde{R}$ saturates to the bound given in Eq. 6 implies*

$$\eta(\tilde{R}) = \left(\frac{F-1}{d^2-1}\right)^{\tilde{R}-1}\frac{F-1}{d(d+1)} + \frac{1}{d}. \tag{8}$$

We leave proof details to Appendix F. A particularly noteworthy aspect of the presented fact is its applicability to general noise models. This allows us to establish a precise relationship between the depth of a quantum circuit $\tilde{R}$, the noise model $\mathcal{E}$, and the quality of approximation (purity) $\eta(\tilde{R}) = 1 - \epsilon$ that can be achieved. This result further implies using measurement operators prepared by pure states suffice to distinguish a weakly noisy quantum state to the maximally mixed state, as defined by Def. 3. In light of this, we consider a noisy quantum state with depth $\tilde{R} = \mathcal{O}(\text{poly}\log n)$, where the noise is modeled as a local depolarizing channel $\mathcal{E}_i(\cdot) = (1-p)(\cdot) + p\text{Tr}_i(\cdot)I_2/2$ and we have $\eta(\tilde{R}) \approx (1 - p\tilde{R})$ by Eq. 8. Furthermore, we note that Eq. 7 implies that there exists a quantum weakly noisy state $\bigcirc_{r=1}^{\tilde{R}}\mathcal{E}\circ\mathcal{U}_r(|0^n\rangle\langle 0^n|)$ and a quantum pure state $|\Psi_{\mathcal{U}_1,\ldots,\mathcal{U}_{\tilde{R}}}\rangle = \mathcal{U}_{\tilde{R}}\cdots\mathcal{U}_1|0^n\rangle$ such that

$$\text{Tr}\left[M_{\mathcal{U}_{\tilde{R}}\cdots\mathcal{U}_1}\left(\bigcirc_{r=1}^{\tilde{R}}\mathcal{E}\circ\mathcal{U}_r(|0^n\rangle\langle 0^n|) - I_n/2^n\right)\right] \approx 1 - p\tilde{R} - 2^{-n} \tag{9}$$

in the worst-case scenario over the choice of $\mathcal{U}_1, \ldots, \mathcal{U}_{\tilde{R}} \sim \mathbb{U}$ (otherwise Eq. 7 may be violated), where the measurement operator $M_{\mathcal{U}_{\tilde{R}}\cdots\mathcal{U}_1} = |\Psi_{\mathcal{U}_1,\ldots,\mathcal{U}_{\tilde{R}}}\rangle\langle\Psi_{\mathcal{U}_1,\ldots,\mathcal{U}_{\tilde{R}}}|$. As a consequence, Definition 1 can be naturally transferred into Definition 3 which characterizes the complexity of quantum weakly noisy states.

## 4 QUANTUM LEARNING TASK AND ALGORITHM

We start by introducing the intrinsic structure of $\mathcal{U}_\mathcal{A}$ to devise a parameterized observable $M$ for distinguishing $\rho$ from the maximal entangled state. In general, optimizing an observable $M = U|0^n\rangle\langle 0^n|U^\dagger$ with $\mathcal{U}_\mathcal{A}(R)$ is challenging, however, the intrinsic structure of a specific QCA allows for efficient optimization of the observable $M$. Specifically, we randomly generate a set of $N = \text{poly}(R, n)$ quantum neural network states $\hat{\rho}_{\text{QCA}}(R, \mathcal{A}, N) = \{\hat{\rho}_i\}_{i=1}^N$ with $\tilde{\rho}_i$ represents the classical shadow representation of $U_i|0^n\rangle$ for $U_i \in \mathcal{U}_\mathcal{A}(R)$ (Huang et al., 2020) (Details refer to Appendix K). We then design a parameterized operator that takes the form of $M(\vec{\beta}) = \sum_{i=1}^N \beta_i\hat{\rho}_i$.

**Theorem 1 (Intrinsic Structure of specific QCA)** *Randomly select $N$ unitaries from the QCA model $\mathcal{U}_\mathcal{A}(R)$ to generate $\hat{\rho}_{\text{QCA}}(R, \mathcal{A}, N) = \{\hat{\rho}_i\}_{i=1}^N$, where each layer in $U_i$ contains $L$ gates. Then for any $n$-qubit quantum state $\rho$ and projector $M_{\vec{x}} = U(\vec{x})|0^n\rangle\langle 0^n|U^\dagger(\vec{x})$ with*

$U(\vec{x}) \in \mathcal{U}_{\mathcal{A}}(R)^5$, *there exists a vector $\vec{\beta}(\vec{x})$ belongs to an $N$-dimensional compact set $\mathcal{D}_{\beta}{}^6$ and $\sum_{j=1}^{N} \vec{\beta}_j(\vec{x}) = 1$, such that*

$$\mathbb{E}_{\vec{x}} \left| \text{Tr} \left[ \sum_{j=1}^{N} \vec{\beta}_j(\vec{x}) \hat{\rho}_i \rho \right] - \text{Tr} \left[ M_{\vec{x}} \rho \right] \right| \leq \sqrt{\frac{LRn^2 \log n}{N}}. \tag{10}$$

*If $N = LRn^2 \log n / \epsilon^2$, the above approximation error is upper bounded by $\epsilon$ (We leave proof details to the Appendix G).*

## 4.1 METRIC CONSTRUCTION

Here, we assume QCA states $\hat{\rho}_i$ are sampled from $\mathcal{U}_{\mathcal{A}}(R)$ following the probability distribution $\vec{q}$, as a result, the metric

$$\mathcal{L}_R(\vec{\beta}) = \left| \mathbb{E}_{\hat{\rho}_i \sim \vec{q}} \text{Tr} \left[ M(\vec{\beta})(\hat{\rho}_i - \rho) \right] \right| \tag{11}$$

is defined over the compact set $\vec{\beta} \in D_{\vec{\beta}}$. Given a specific parameter $\vec{\beta}$, we can efficiently calculate the corresponding value of $\mathcal{L}_R(\vec{\beta})$ using classical shadow techniques (Huang et al., 2020; Wu et al., 2023a; Nguyen et al., 2022; Akhtar et al., 2022; Bertoni et al., 2022).

To clarify our algorithm, we define the upper decision interval $\text{UDI}(\epsilon)$ and the lower decision interval $\text{LDI}(\epsilon)$ as follows.

**Definition 4 (Decision interval)** $\text{UDI}(\epsilon)$ *is defined as the integer interval $[u, \infty)$ such that for any $R \in \text{UDI}(\epsilon)$, it holds that $\max_{\vec{\beta}} \mathcal{L}_R \left( \vec{\beta} \right) \leq \epsilon$. Similarly, $\text{LDI}(\epsilon)$ is defined as the integer interval $[0, l]$ such that for any $R \in \text{LDI}(\epsilon)$, we have $\min_{\vec{\beta}} \mathcal{L}_R \left( \vec{\beta} \right) \geq 2\epsilon$.*

Before proposing the quantum learning algorithm, we need the following lemmas to support our method. We elaborate proof details in Appendices H.1 and H.2.

**Lemma 1** *Consider the metric function $\mathcal{L}_R(\vec{\beta})$. If the relationship $\max_{\vec{\beta}} \mathcal{L}_R(\vec{\beta}) \leq \epsilon$ holds for any distribution $\vec{q}$, then $\rho_{\text{un}}$ has the state complexity $C_{\epsilon}^{\lim,\mathcal{A}}(\rho_{\text{un}}) \leq LR$, where $C_{\epsilon}^{\lim,\mathcal{A}}(\cdot)$ is defined in Def. 3.*

**Lemma 2** *If there exists a distribution $\vec{q}$ such that $\min_{\vec{\beta}} \mathcal{L}_R(\vec{\beta}) > 2\epsilon$, then with nearly unit probability, $\rho_{\text{un}}$ follows the quantum state complexity lower bound $C_{\epsilon}^{\lim,\mathcal{A}}(\rho_{\text{un}}) > LR$.*

## 4.2 LEARNING TASK STATEMENT

Given the preliminary background above, we now formally define the learning task.

**Task 1 (Structured Complexity Prediction ($\text{SCP}(\mathcal{A}, \epsilon)$))** *Given an architecture $\mathcal{A}$, an $n$-qubit weakly noisy quantum state $\rho_{\text{un}}$ (as defined in Eq. 4), and an approximation error $\epsilon$, design a learning algorithm $\text{QL}_n$ that runs on an ideal quantum device with polynomially many qubits in $n$, and learns from the unknown quantum state $\rho_{\text{un}}$, such that the following conditions hold:*

1. *(**Completeness**) If there exists an integer $x < \log n$ such that $x \in \text{UDI}(\epsilon)$, then $\text{QL}_n$ returns `True`.*

2. *(**Soundness**) If $\log n \in \text{LDI}(\epsilon)$, then $\text{QL}_n$ returns `False`.*

3. *(**Indeterminate case**) If all integers $x \in [0, \log n]$ lie in $\text{UDI}^c(\epsilon) \cup \text{LDI}^c(\epsilon)$, then $\text{QL}_n$ may return either `True` or `False` arbitrarily.*

---

[5] Here, the vector $\vec{x} \in [0, 2\pi]^{LR}$ uniquely determines a quantum circuit $U(\vec{x}) \in \mathcal{U}_{\mathcal{A}}(R)$.

[6] To keep the semi-definite property of $M(\vec{\beta})$, we generally assume the compact set $\mathcal{D}_{\beta} = [0, 1]^N$.

To provide a clearer analysis of this task, we introduce the following definition.

**Definition 5 (Distinguishable Property, $\mathrm{DP}(\mathcal{A}, \epsilon)$)** *We say an $n$-qubit weakly noisy quantum state $\rho_{\mathrm{un}}$ satisfies $\mathrm{DP}(\mathcal{A}, \epsilon)$ if $\rho_{\mathrm{un}}$ can be distinguished from $I_n/2^n$ with the bias at least $1 - 2^{-n} - \epsilon$ by using a $R$-depth ($R < \log n$) QCA measurement $M_{\mathrm{QCA}} = U|0^n\rangle\langle 0^n|U^\dagger$, where $U \in \mathcal{U}_{\mathcal{A}}$.*

We note that the learning algorithm $\mathrm{QL}_n$ returns `True` (respectively, `False`) to indicate that $\rho_{\mathrm{un}}$ does (does not) satisfy the $\mathrm{DP}(\mathcal{A}, \epsilon)$ property.

### 4.3 Quantum Learning Algorithm for Limited-Structured Complexity Analysis

Based on Theorem 1, optimizing the metric function $\mathcal{L}_R(\vec{\beta})$ can be limited into a compact set. Taking the maximization task as an example, we show how to utilize the Bayesian optimization on a compact set as Alg. 2 in Appendix I, which is termed as the BMaxS. Specifically, the subroutine BMaxS, *which is essentially a Bayesian optimization*, takes unknown state $\rho_{\mathrm{un}}$, QCA state set, and LS-error $\epsilon$ as inputs, after $T$ steps, it outputs `True` if $R \in \mathrm{UDI}(\epsilon)$ holds, outputs `False` otherwise. This subroutine can be used as a building block to construct a quantum learning algorithm that verifies whether the completeness condition of Task 1 holds. Likewise, BMaxS can also be applied to verify the soundness condition. For the remaining (inconclusive) case, we return a random outcome (`True` or `False`), thereby providing an efficient solution to Task 1.

Given the fact that a structured unitary set that $\mathcal{U}_{\mathcal{A}}(R-1)$ is strictly contained in $\mathcal{U}_{\mathcal{A}}(R)$ (Haferkamp et al., 2022), the boolean function

$$\mathcal{P}(R, \epsilon) = \mathrm{BMaxS}(\rho_{\mathrm{un}}, \hat{\rho}_{\mathrm{QCA}}(R, \mathcal{A}, N), T, \epsilon) \tag{12}$$

is a monotone predicate in the interval $R \in [1, \log n]$. Therefore a quantum learning algorithm can be designed by a binary search program where $\mathcal{P}(R, \epsilon)$ is packaged as an oracle. A monotone predicate $\mathcal{P}$ is a boolean function defined on a totally ordered set with the property: if $\mathcal{P}(R, \epsilon) = $ `True`, then $\mathcal{P}(R', \epsilon) = $ `True` for all $R' \geq R$ in the domain. In our case, $\mathcal{P}$ returns `True` for the input $R$ when $R \in \mathrm{UDI}(\epsilon)$ holds. As a result, if the noisy state $\rho_{\mathrm{un}}$ satisfies the $\mathrm{DP}(\mathcal{A}, \epsilon)$ property, the $\mathrm{QL}_n$ outputs the minimum $R_{\min} \in [1, \log n]$ enabling $C_\epsilon^{\mathrm{lim}, \mathcal{A}}(\rho_{\mathrm{un}}) \leq LR_{\min}$ (`True`). Otherwise, we test whether $\log n \in \mathrm{LDI}(\epsilon)$. If this holds, the $\mathrm{QL}_n$ outputs $C_\epsilon^{\mathrm{lim}, \mathcal{A}}(\rho_{\mathrm{un}}) > L \log n$ (`False`). Otherwise, the studied $\rho_{\mathrm{un}}$ and threshold $\epsilon$ would be an invalid case. Details are provided in Alg. 1.

---

**Algorithm 1:** Quantum Learning Algorithm for Limited-Structured Complexity Prediction

---

**Input** : Noisy quantum state $\rho_{\mathrm{un}}$, $\epsilon$;
**Output:** The minimum depth $R$ ($R < \log n$) such that $C_\epsilon^{\mathrm{lim}, \mathcal{A}}(\rho_{\mathrm{un}}) \leq LR$ (True);
False if such $R$ does not exist; Or return True/False arbitrarily for invalid cases;

1  **Initialize** $R \leftarrow 1$, $s \leftarrow \log(n)$;
2  **while** $s - R > 1$ **do**
3     Set $N = LRn^2 \log n/\epsilon^2$ and $T = N^2 n^k$ such that $k \log(n) < n^{k/2-1}\epsilon$ for large $n$;
4     **if** $\mathcal{P}((R+s)/2, \epsilon) = $`True` **do**
5         $s \leftarrow \lceil (R+s)/2 \rceil$
6     **else do**
7         $R \leftarrow \lceil (R+s)/2 \rceil$
8  **if** $\mathcal{P}(R, \epsilon) = $`True` **do**
9     **return** $C_\epsilon^{\mathrm{lim}, \mathcal{A}}(\rho_{\mathrm{un}}) \leq LR$ *(True)*
10 **else do**
11     **if** $\mathcal{P}(R, 2\epsilon) = $`False`, **return** $C_\epsilon^{\mathrm{lim}, \mathcal{A}}(\rho_{\mathrm{un}}) > L \log n$ *(False)*
12     **else do**
13         **return** *True or False arbitrarily*

---

## 5 Theoretical Performance Guarantee

We will demonstrate that the required number of samplings and processing time for Alg. 1 are both efficient, and the sample complexity is optimal with respect to the noisy circuit depth. Using Theorem 1, we can state the following result.

**Theorem 2** *Given* $\text{poly}(n)$ *copies of an* $n$*-qubit unknown weakly noisy state* $\rho_{\text{un}}$ *that is generated by a noisy quantum device with depth* $\tilde{R} = \mathcal{O}(\log n)$ *(Def. 2) and a particular architecture* $\mathcal{A}$*, there exists a* $\text{poly}(n, \tilde{R}, 1/\epsilon)$ *quantum and classical cost learning algorithm which can efficiently solve the* $\text{SCP}(\mathcal{A}, \epsilon)$ *problem.*

The proof of Theorem 2 depends on evaluating the sample complexity of QCA states and unknown noisy state, as well as the related iteration complexity during training the quantum learning algorithm. We analyze the sample and classical computational complexity in the next two subsections as part of the proof outline for this theorem.

## 5.1 SAMPLE COMPLEXITY

The sample complexity of QCA states is promised by Theorem 1, where the number of copies of QCA states $N = LRn^2 \log n/\epsilon^2$. To evaluate $\mathcal{L}_R(\vec{\beta})$ for $1 \leq R \leq \tilde{R} \leq \mathcal{O}(\log n)$ within $\epsilon$-additive error, the classical shadow representation of $\rho_{\text{un}}$ is shared by all QCA states, by leveraging the classical shadow method (Huang et al., 2020). As a result, the sample complexity of an unknown weakly noisy state is at most $m \leq \mathcal{O}\left(\frac{n^2 L \tilde{R} \log n \log(1/\delta)}{\epsilon^4}\right)$, where the failure probability $\delta \in (0, 1)$.

Meanwhile, we also give a sample complexity lower bound to demonstrate our learning algorithm is optimal in terms of the circuit depth $\tilde{R}$.

**Theorem 3** *Given an unknown weakly noisy quantum state* $\rho$ *prepared by a* $R$*-depth quantum circuit affected by* $p$*-strength local Pauli noise channels, then any algorithm designed to learn* $\rho$ *requires at least* $m$ *samplings in the worst-case scenario, where* $m = \frac{(1-p)^{-2c\tilde{R}}(1-\delta)^2}{2n}$*,* $c = 1/(2 \ln 2)$ *and constant* $\delta \in \mathcal{O}(1)$*.*

From the above result and the definition of the quantum weakly noisy state, we know that $p = \mathcal{O}(n^{-1})$ which results in the sample complexity lower bound $\Omega((1 + 2cp\tilde{R})(1 - \delta)^2/(2n))$. This implies the sample complexity lower bound may linearly growth with the increasing of the circuit layer $\tilde{R}$, and our learning algorithm is nearly-optimal in terms of the circuit depth $\tilde{R}$. We leave proof details to Appendix J.

## 5.2 RUNNING TIME ANALYSIS

Here, we provide an analysis of the computational complexity for Alg. 1, including the number of iterations in Alg. 1 and the subroutine BMaxS (Alg. 2) which is a Bayesian optimization algorithm. Denote the global optimum $\vec{\beta}^* = \max_{\vec{\beta} \in \mathcal{D}_{\vec{\beta}}} \mathcal{L}_R(\vec{\beta})$, and a natural performance metric for optimizing $\mathcal{P}(R)$ is the *simple regret* $s_T = \mathcal{L}_R(\vec{\beta}^*) - \mathcal{L}_R(\vec{\beta}^{(T)})$, which is the difference between the global maximum $\mathcal{L}_R(\vec{\beta}^*)$ and $\mathcal{L}_R(\vec{\beta}^{(T)})$. Here, $\vec{\beta}^{(T)}$ represents the updated parameter in the $T$-th step. Obviously, simple regret is non-negative and asymptotically decreases with the increasing iteration complexity $T$. To build up an explicit connection between $s_T$ and $T$, the *average regret* $\text{avr}_T$ is introduced. Specifically, $\text{avr}_T = 1/T \sum_{t=1}^{T} \left[\mathcal{L}_R(\vec{\beta}^*) - \mathcal{L}_R(\vec{\beta}^{(T)})\right]$. Noting that the relationship $s_T \leq \text{avr}_T$ holds for any $T \geq 1$. In the following, we show that $\text{avr}_T$ is upper bounded by $\mathcal{O}(N \log T/\sqrt{T})$, and the simple regret $s_T \leq \text{avr}_T \to 0$ with the increase of $T$. The following theorem derives the average regret bounds for $\mathcal{P}(R) = \text{BMaxS}(\rho_{\text{un}}, \hat{\rho}_{\text{QCA}}(R, \mathcal{A}, N), T, \epsilon)$.

**Theorem 4** *Take the target weakly noisy state* $\rho_{\text{un}}$ *and QCA state set* $\hat{\rho}_{\text{QCA}}(R, \mathcal{A}, N)$ *into the subroutine* $\mathcal{P}(R)$ *(Eq. 12). Pick the failure probability* $\delta \in (0, 1)$*, then there exists a Bayesian approach (details refer to Alg. 2) such that the average regret* $\text{avr}_T$ *can be upper bounded by*

$$\text{avr}_T \leq \mathcal{O}\left(\sqrt{\frac{4N^2 \log^2 T + 2N \log T \log(\pi^2/(6\delta))}{T}}\right) \tag{13}$$

*after* $T$ *optimization steps with* $1 - \delta$ *success probability, where* $N$ *represents the number of samples in the QCA state set.*

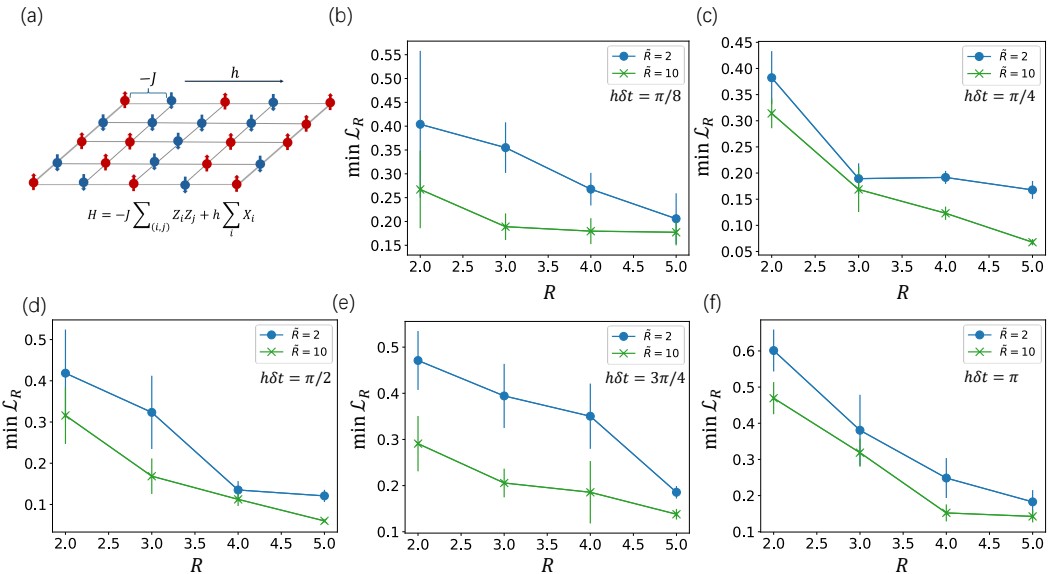

Figure 2: (a) Visualization of the 2D transverse field Ising model. (b)-(d) illustrate the trend of the function $\min_{\vec{\beta}} \mathcal{L}_R$ as it varies with the circuit depth $R$ of QCA set. To extract the complexity lower bound of the target quantum state from the plot, we begin by drawing a horizontal line representing the approximation error of the quantum state. For example, in (b), we set $\epsilon' = 0.25$. Next, we identify the last point where the curve remains above this horizontal line before crossing it, and record the corresponding horizontal coordinate $R$. This $R$ represents the complexity lower bound of the target quantum state. For instance, in Fig. (b), the blue curve remains above $\epsilon' = 0.25$ until $R = 4$, indicating a complexity lower bound of $C_{0.125}^{\lim,\mathcal{A}}(\rho_{\text{TI}}(2,p)) > 4L$. Similarly, the green curve last remains above $\epsilon' = 0.25$ at $R = 2$, yielding a complexity lower bound of $C_{0.125}^{\lim,\mathcal{A}}(\rho_{\text{TI}}(10,p)) > 2L$, where $L$ represents the number of two-qubit gates in each layer of the architecture $\mathcal{A}$.

Specifically, select an integer $k$ such that $k \log(n) < n^{k/2}\epsilon$ for all $n > n_0$, where $n_0$ represents a fixed integer. Then $T = N^2 n^k$ enables the simple regret $s_T$ to be upper bounded by $\epsilon$, where $N = LRn^2 \log n\epsilon^{-2}$ (Theorem 1). The proof details refer to Appendix I.

Finally, noting that Alg. 1 is essentially a binary search program on the interval $[1, \tilde{R}]$ by using the oracle $\mathcal{P}(R, \epsilon) = \text{BMaxS}(\rho_{\text{un}}, \hat{\rho}_{\text{QCA}}(R, \mathcal{A}, N), T, \epsilon)$. Therefore, Alg. 1 requires

$$\mathcal{O}\left(\frac{1}{\epsilon^4}, n^{4+k}, L^2, \tilde{R}^2 \log(\tilde{R}), \log(1/\delta)\right)$$

classical time complexity to answer the SCP problem. This thus completes the proof of Theorem 2.

## 6 NUMERICAL SIMULATIONS

Here, we demonstrate how to use the proposed learning method to benchmark the capabilities of noisy state computation, providing numerical evidence to support our theoretical findings. Specifically, we address the fundamental question: *Does the complexity of weakly noisy quantum states grow linearly with circuit depth?*

We consider to simulate the time dynamics of the Hamiltonian $H = -J \sum_{\langle i,j \rangle} Z_i Z_j + h \sum_i X_i$ on a two-dimensional grid with $(a \times b)$ size, where $J > 0$ represents the coupling of nearest-neighbour spins and $h$ represents the global transverse field strength (see Fig. 2 (a)). To simulate the time evolution circuit $e^{-iH\tau}$, the first-order Trotter decomposition is utilized (Kim et al., 2023), that is $e^{-iH\tau} = \left[e^{-iH_{\text{ZZ}}\delta t}e^{-iH_{\text{X}}\delta t}\right]^{\tau/\delta t} + \mathcal{O}((\delta t)^2)$, where $H_{\text{ZZ}}$ represents the spin term, $H_{\text{X}}$ represents the transverse-field term and the evolution time $\tau$ is discretized into $(\tau/\delta t)$ time slices. Then its quantum circuit implementation can be decomposed by $e^{-iH_{\text{ZZ}}\delta t} = \prod_{\langle i,j \rangle} R_{Z_i Z_j}(-2J\delta t)$ and

$e^{-iH_X \delta t} = \prod_i R_{X_i}(2h\delta t)$. We focus on a small-scale scenario studied in Ref. (Kim et al., 2023), where the system size is $3 \times 4$, angle rotations $-2J\delta t = -\pi/2$, $h\delta t \in \{\pi/8, \pi/4, \pi/2, 3\pi/4, \pi\}$ and each quantum gate is affected by a local depolarizing channel $\mathcal{E}_i$ with strength $p = 10^{-3}$. Let $\tilde{R} = \lceil \tau/\delta t \rceil$, then we denote the output quantum weakly noisy state as

$$\rho_{\mathrm{TI}}(\tilde{R}, p) = \bigcirc_{r=1}^{\tilde{R}} \left[ \mathcal{E}_p \circ \mathcal{U}_{ZZ} \circ \mathcal{E}_p \circ \mathcal{U}_X \right] (|0^n\rangle\langle 0^n|), \tag{14}$$

where $\mathcal{U}_{ZZ}(\cdot) = e^{-iH_{ZZ}\delta t}(\cdot)e^{iH_{ZZ}\delta t}$, $\mathcal{U}_X(\cdot) = e^{-iH_X\delta t}(\cdot)e^{iH_X\delta t}$ and the quantum noise channel $\mathcal{E}_p = \otimes_{i=1}^n \mathcal{E}_i$. In the numerical simulation, we demonstrated our algorithm on a server with 64 vCPUs and 128 GiB of memory, where the density matrix $\rho_{\mathrm{TI}}(\tilde{R}, p)$ and classical shadow set are prepared by the Pennylane package (Bergholm et al., 2018).

In our analysis, we mainly focus on the *circuit complexity lower bound* of noisy quantum states $\rho_{\mathrm{TI}}(\tilde{R}, p)$ with $\tilde{R} \in \{2, 10\}$ by using Alg. 1, considering a standard 2D lattice quantum circuit architecture $\mathcal{A}$ to prepare $\hat{\rho}_{\mathrm{QCA}}$, where each layer has $L = \mathcal{O}(n)$ random two-qubit gates. Visualization of the architecture $\mathcal{A}$ in 1D scenario is given by Fig. 1 (b). To estimate the lower bound, we thus set a small error $(\epsilon = 10^{-2})$[7] and randomly generate $N = n^2 R$ quantum circuits with varying circuit depths $R \in \{2, 3, 4, 5\}$ based on the architecture $\mathcal{A}$. Precisely, we tune the linear coefficient $\vec{\beta}$ to minimize the metric functions outlined in Lemma 2, as depicted in Figure 2, where each point represents the mean value of $\min_{\vec{\beta}} \mathcal{L}_R$ by repeating 10 independent experiments, and the error bar represents the standard variance. This strategic adjustment allows us to derive an estimate for the quantum circuit lower bound. More specifically, in Figure 2 (b), we showcase a clear circuit complexity separation between weakly noisy quantum states $\rho_{\mathrm{TI}}(2, p)$ and $\rho_{\mathrm{TI}}(10, p)$. To see this relationship, Figure 2 (b) demonstrated that $\min_{\vec{\beta}} \mathcal{L}_2(\rho_{\mathrm{TI}}(10, p)) > 0.25$, meanwhile $\min_{\vec{\beta}} \mathcal{L}_4(\rho_{\mathrm{TI}}(2, p)) > 0.25$ which demonstrate that the circuit complexity $C_{0.125}^{\mathrm{lim}, \mathcal{A}}(\rho_{\mathrm{TI}}(\tilde{R} = 10, p)) > 2L$ and $C_{0.125}^{\mathrm{lim}, \mathcal{A}}(\rho_{\mathrm{TI}}(\tilde{R} = 2, p)) > 4L$ (according to Lemma 2), where $L = 3n$ represents the number of random two qubit gates in each layer. *This result highlights weakly noisy quantum state complexity lower bound may not grow with the circuit depth, which is dramatically different to that of pure states.* Similar phenomenons are witnessed in subfigures. 2 (c)-(f), where a shallower circuit depth weakly noisy states possess higher state complexity lower bound.

## 7 CONCLUSION

The quantum state complexity serves as a measure of inherent properties within quantum states, thereby facilitating a deeper understanding of quantum entanglement information, quantum topological phases, and computational capabilities. In practical applications, collected quantum states are often subject to noise originating from state preparation and quantum measurement (SPAM), as well as limitations imposed by the quantum hardware. Consequently, original pure states are transformed to noisy states through quantum channels. Thus, investigating the quantum state complexity of noisy states holds significant importance in studying information scrambling, the spread of local noise and entanglement throughout the entire system, which is expected to illuminate studies in the field of black-hole theory and condensed-matter physics. In this paper, we investigate the complexity of weakly noisy quantum states through a quantum learning algorithm, which connects two significant concepts in the quantum computational theory. The proposed quantum learning algorithm exploits the intrinsic structure of QCA to build a learning model $\mathcal{L}_R(\beta)$, whose extreme points reveal the limited-structured complexity. Meanwhile, when considering the sample complexity of target noisy quantum state, our algorithm achieves optimal in terms of the circuit depth $\tilde{R}$. Moreover, we emphasize that the Bayesian optimizer (given by Alg. 2) is not the unique option, and other optimization algorithms may also work with a similar iteration steps. This highlights the universality of the intrinsic structure of QCA in combination with optimization subroutines.

---

[7]Here, the numerical simulation mainly aims to find the circuit lower bound. Therefore, we set a small error threshold $\epsilon$ such that the measurement operators constructed in Alg 1 cannot lead to $\mathcal{P}(R) = \mathrm{true}$. In this case, the algorithm will proceed to the final step to determine the lower bound of the quantum state's complexity.

ACKNOWLEDGMENTS

We thank Jens Eisert, Philippe Faist, Haihan Wu, Tavis Bennett, John Tanner, and all anonymous reviewers for their helpful comments on this paper. Y. Wu acknowledges the support from the Natural Science Foundation of China Grant (No. 62371050). B. Wu acknowledges the support from the National Natural Science Foundation of China Grant (No. 12405014) and the Research fund of the post-doctor who came to Shenzhen. X. Yuan is supported by the Innovation Program for Quantum Science and Technology Grant (No. 2023ZD0300200), the National Natural Science Foundation of China Grant (No. 12175003 and No. 12361161602), NSAF Grant (No. U2330201). J. B. Wang acknowledges continued support from Pawsey Supercomputing Research Centre.

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

## A    COMPARISON WITH RELATED WORKS

In the context of condensed matter physics, a topological phase transition occurs when the ground states of a family of Hamiltonians exhibit different circuit complexities. This suggests that topological phase classification may help distinguish between low-complexity and high-complexity ground states. The use of learning algorithms to classify quantum phases of matter has been widely studied. Proposals include quantum neural networks (Cong et al., 2019), classical neural networks (Beach et al., 2018; Carrasquilla & Melko, 2017; Greplova et al., 2020; Schindler et al., 2017; Van Nieuwenburg et al., 2017), and other classical machine learning models (Huang et al., 2022; Rodriguez-Nieva & Scheurer, 2019; Wetzel, 2017). However, most previous works lack rigorous theoretical guarantees. Among these studies, Huang et al. (Huang et al., 2022) utilized shadow tomography of the concerned ground states to design an unsupervised machine learning approach guaranteed to classify accurately under certain conditions. The proposed "shadow kernel" can generate quantum entropies by tuning hyperparameters, enabling the approximation of various topological phase order parameters. Compared with Ref. Huang et al. (2022), our learning algorithm can be applied to noisy states and predict complexity, whereas Ref. Huang et al. (2022) mainly focuses on pure ground state classification without providing an explicit metric for characterizing complexity.

On the other hand, learning quantum states and circuits is a long-standing task in the field of quantum machine learning. Previous works generally utilized parameterized circuits to learn approximations of target states and circuits (Mitarai et al., 2018; Yang et al., 2020; Huang et al., 2021; Jerbi et al., 2021; Cerezo et al., 2022). However, variational-based methods generally lack theoretical guarantees and may suffer from the barren plateau phenomenon even in low-depth parameterized circuits (McClean et al., 2018; Cerezo et al., 2021; Anschuetz & Kiani, 2022). Very recently, classical shadow-based learning algorithms have been proposed for reconstructing shallow quantum circuits $U$ (Huang et al., 2024; Landau & Liu, 2024). Nevertheless, these methods require querying $U^\dagger$, which introduces intrinsic limitations in noisy environments since most noisy channels (CPTP maps) may not be invertible. Compared with related works, our method bypasses these limitations by introducing the "Intrinsic Structure property" (as given by Theorem 1), which enables an efficient quantum learning algorithm to predict state complexity in noisy environments.

we would like to point out that Ref. Schuster et al. (2024) does not rule out the possibility that the pure state complexity can be predicted when the target states exhibit certain physical structures, such as the ground state of XXZ models and the toric code, as studied in Ref. Huang et al. (2022). Furthermore, recent works (Huang et al., 2024; Landau & Liu, 2024) have also demonstrated that fixed shallow quantum circuits and quantum pure states (including 1D-$\mathcal{O}(\log n)$-depth and all-to-all $\mathcal{O}(\log \log n)$-depth) can be efficiently learned. These results demonstrate significant differences between scenarios with and without 'randomness'. Compared with Ref. (Schuster et al., 2024), our work is more similar to Refs. (Huang et al., 2022; 2024; Landau & Liu, 2024), which focus on a specific quantum state rather than an ensemble.

Furthermore, we would like to highlight the differences between weakly noisy quantum states and pseudorandom quantum states. Specifically, we consider the second-moment statistic property of a set of pseudorandom quantum circuits $U$ whose circuit depth $\tilde{R} \leq \text{poly} \log n$. In Ref. (Schuster et al., 2024), they claimed that even $\mathcal{O}(\log(n))$-depth Clifford circuits may approximate the unitary 2-design property given by Haar measure. Then, we may suppose $U = U_1 \cdots U_{\tilde{R}}$ representing a random Clifford circuit, then for the initial state $|0^n\rangle\langle 0^n|$ and arbitrary observable $\hat{O}$, we have

$$M_2(U) = \int_{U \sim \text{Cl}(2^n)} \text{Tr}\left[U|0^n\rangle\langle 0^n|U^\dagger O\right] \text{Tr}\left[U|0^n\rangle\langle 0^n|U^\dagger O\right] = \frac{1}{d(d^2-1)}\left[\text{Tr}^2[O] + \text{Tr}[O^2]\right]. \tag{15}$$

On the other hand, in the context of weakly noisy environment, the Clifford circuit $U$ may transform to the channel representation $\mathcal{U} = \bigcirc_{r=1}^{\tilde{R}} \mathcal{E} \circ \mathcal{U}_r$ where $\mathcal{E}$ represents a $n$-qubit Pauli channel and $\mathcal{U}_r = U_r(\cdot)U_r^\dagger$ represents a layer of Clifford gate. According to the "Channel Pushing" lemma given by Ref Quek et al. (2024) (Lemma 10), we may rewrite the noisy channel by $\mathcal{U} = \mathcal{E}' \circ \bigcirc_{r=1}^{\tilde{R}} \mathcal{U}_r$, where $\mathcal{E}'$ also represents an $n$-qubit Pauli channel. Let the channel $\mathcal{E}' = \sum_l K_l(\cdot)K_l^\dagger$, where $K_l$

represents the Kraus operator satisfying $\sum_l K_l^\dagger K_l = I$, we have

$$
\begin{aligned}
M_2(\mathcal{U}) &= \int_{U \sim \mathrm{Cl}(2^n)} \mathrm{Tr}\left[\mathcal{U}(|0^n\rangle\langle 0^n|)O\right] \mathrm{Tr}\left[\mathcal{U}(|0^n\rangle\langle 0^n|)O\right] \\
&= \sum_{l_1,l_2} \int_{U \sim \mathrm{Cl}(2^n)} \mathrm{Tr}\left[K_{l_1} U(|0^n\rangle\langle 0^n|)U^\dagger K_{l_1}^\dagger O\right] \mathrm{Tr}\left[K_{l_2} U(|0^n\rangle\langle 0^n|)U^\dagger K_{l_2}^\dagger O\right] \\
&= \sum_{l_1,l_2} \left(\frac{1}{d(d^2-1)}\left(\mathrm{Tr}(K_{l_1} O K_{l_1}^\dagger)\mathrm{Tr}(K_{l_2} O K_{l_2}^\dagger) + \mathrm{Tr}(K_{l_1} O K_{l_1}^\dagger K_{l_2} O K_{l_2}^\dagger)\right)\right).
\end{aligned}
\tag{16}
$$

We note that the interaction term $\sum_{l_1,l_2} \mathrm{Tr}(K_{l_1} O K_{l_1}^\dagger K_{l_2} O K_{l_2}^\dagger)$ may not equal to $\mathrm{Tr}(O^2)$, and this leads to $M_2(U) \neq M_2(\mathcal{U})$. This comparison indicates that *even if an algorithm can learn the difference between noisy ensembles and Haar-random states, this does not imply that the algorithm can be used to distinguish between pseudorandom ensembles and Haar-random ensembles.*

The above discussions demonstrate the fundamental difference between our work and Ref. (Schuster et al., 2024). Such difference implies that the efficiency of our learning algorithm may not contradict the findings presented in Ref. (Schuster et al., 2024).

## B   NOISE MODELS ASSUMPTION IN THIS WORK

In the proposed learning approach (Alg 1), the algorithm requires (i) the target weakly noisy quantum state $\rho_{\mathrm{un}}$, and (ii) the QCA state set $\hat{\rho}_{\mathrm{QCA}}$. The target weakly noisy state naturally contains noise signals, while the QCA state set $\hat{\rho}_{\mathrm{QCA}}$ is provided by the classical shadow representation (Huang et al., 2020). When preparing the classical shadow, recent result successfully embedded the quantum error mitigation method into the classical shadow protocol, which provides rigorous theoretical guarantees even in the noisy environment (Jnane et al., 2024).

In quantum computing research, the gate-independent noise model assumes that the noise affecting quantum operations is uniform across all gates, regardless of their type or implementation. This simplification is widely adopted for several reasons:

> **Theoretical Simplification:** Assuming gate-independent noise allows researchers to develop and analyze error correction protocols and fault-tolerant methods without delving into the complexities introduced by gate-specific noise characteristics. This uniformity facilitates the derivation of general results and theoretical bounds (Nielsen & Chuang, 2001).

> **Practical Approximations:** In certain quantum systems, particularly those with well-calibrated gates operating on the same number of qubits and utilizing uniform control mechanisms, noise variations across different gates can be negligible (Shor, 1996; Arute et al., 2019). In such cases, the gate-independent noise model serves as a reasonable approximation, streamlining analysis without significantly compromising accuracy.

> **Alignment with Twirled Noise Models:** Techniques like Pauli twirling are employed to transform complex noise channels into diagonal forms on the Pauli basis (Chen et al., 2023). While twirling simplifies the noise structure, it does not inherently eliminate gate dependence. However, in many scenarios, the resulting noise can be approximated as gate-independent, aligning with the assumptions of the model.

The gate-independent noise model provides a foundational framework for understanding error propagation and developing correction strategies, which is a useful abstraction for theoretical exploration and the initial development of error correction methods. We leave an open question of how to depict the gate-dependent noise, which usually happens in larger, more complex quantum architectures.

## C   QUANTUM HARDWARE REQUIREMENT

We note that our learning algorithm has very few limitations to the practical quantum hardware. In the Alg. 1, we only require (i) the target weakly noisy quantum state $\rho_{\mathrm{un}}$, and (ii) the QCA state set $\hat{\rho}_{\mathrm{QCA}}$, which is essentially the classical shadow representation Huang et al. (2020). When preparing

the classical shadow, recent work has successfully integrated quantum error mitigation methods into the classical shadow protocol, providing rigorous theoretical guarantees in preparing shadows even in noisy environments Jnane et al. (2024). As a result, our Alg 1 only require a quantum computer which supports the quantum error mitigation function, which should have $50 - 100$ noisy qubits with $\geq 0.99$ quantum gate fidelity. Some popular quantum computation platforms may satisfy these requirements and can be used to prepare the required classical shadow, such as Eagle used in Ref. Kim et al. (2023) and Sycamore Arute et al. (2019). This work is essentially a *theoretical research*, and the algorithm has not yet been explicitly tested on these practical platforms.

## D  COMPARISON BETWEEN LOCAL AND GLOBAL NOISE MODELS

We note that the global depolarizing channel is essentially a special case of contracting $n$ local noise models. Specifically, given the global depolarizing channel $\mathcal{E}$ with noisy strength $p_g$ and local depolarizing channel $\otimes_j \mathcal{E}_j$ with strength $p_l$, according to the relative entropy inequality, for any input state $\rho$, we can observe the relationships

$$D(\mathcal{E}(\rho)\|I_n/2^n) \leq (1 - p_g)D(\rho\|I_n/2^n), \tag{17}$$

and

$$D(\otimes_{j=1}^n \mathcal{E}_j(\rho)\|I_n/2^n) \leq (1 - p_l)^n D(\rho\|I_n/2^n) \tag{18}$$

hold. These inequalities indicate that the global depolarizing channel is much weaker than local noise, given the same noise strength. Consequently, for the global depolarizing channel, our algorithm is capable of handling scenarios where $p_g \leq \mathcal{O}(1)$, while maintaining comparable prediction accuracy. This suggests that the algorithm can effectively handle both types of noise, with only a minor difference in performance under different noise strengths.

## E  RELATED DEFINITIONS

**Definition 6 (Architecture Haferkamp et al. (2022); Bouland et al. (2019a))** *An architecture $\mathcal{A}$ is a directed acyclic graph that contains $|\mathcal{V}| \in \mathbb{Z}_{>0}$ vertices (gates), and two edges (qubits) enter each vertex, and two edges exit. A quantum circuit induced by the architecture $\mathcal{A}$ is denoted as $U_{\mathcal{A}}$. The circuit set which contains all quantum circuits induced by the architecture $\mathcal{A}$ is denoted by $\mathcal{U}_{\mathcal{A}}$. The circuit set $\mathcal{U}_{\mathcal{A}}(R)$ contains all $R$-depth quantum circuits induced by the architecture $\mathcal{A}$, and we have the relationship*

$$\mathcal{U}_{\mathcal{A}} = \cup_{R \geq 1} \mathcal{U}_{\mathcal{A}}(R). \tag{19}$$

**Definition 7 (Causal Slice)** *The circuit $U_{\mathcal{A}}$ is a causal slice if there exists a qubit-reachable path between any two qubit-pairs, where the path only passes through vertices (gates) in the architecture $\mathcal{A}$.*

## F  PROOF OF FACT 1

Here, we are interested in measuring the quantity $\mathbb{E}_{\mathcal{U}_1,\ldots,\mathcal{U}_{\tilde{R}}}\left[\text{Tr}(\rho_{1,\tilde{R}}\rho_{2,\tilde{R}})\right]$, where $\rho_{1,\tilde{R}} = \bigcirc_{r=1}^{\tilde{R}} \mathcal{E} \circ \mathcal{U}_r(\rho_1)$, $\rho_{2,\tilde{R}} = \bigcirc_{r=1}^{\tilde{R}} \mathcal{U}_r(\rho_2)$ and $\rho_1$, $\rho_2$ represent initial pure states. Suppose $\mathcal{E}$ can be decomposed by Kraus operators, that is $\mathcal{E}(\cdot) = \sum_{l=1}^r K_l(\cdot)K_l^{\dagger}$. Then we first consider the scenario $\tilde{R} = 1$:

$$\begin{aligned}
\text{Tr}\left(\rho_{1,1}\rho_{2,1}\right) &= \text{Tr}\left[(\mathcal{E} \circ \mathcal{U}_1(\rho_1)\mathcal{U}_1(\rho_2))\right] \\
&= \text{Tr}\left[\text{Swap}\left(\mathcal{E} \circ \mathcal{U}_1(\rho_1) \otimes \mathcal{U}_1(\rho_2)\right)\right] \\
&= \sum_{l=1}^r \text{Tr}\left[\text{Swap}\left(K_l\mathcal{U}_1(\rho_1)K_l^{\dagger} \otimes \mathcal{U}_1(\rho_2)\right)\right].
\end{aligned} \tag{20}$$

Then taking the average value on a unitary 2 design set, we obtain

$$\mathbb{E}_{\mathcal{U}_1} \text{Tr}\,(\rho_{1,1}\rho_{2,1}) = \sum_{l=1}^{r} \mathbb{E}_{\mathcal{U}_1} \text{Tr}\left[\text{Swap}\left((K_l \otimes I_n)(U_1 \otimes U_1)(\rho_1 \otimes \rho_2)(U_1^\dagger \otimes U_1^\dagger)(K_l^\dagger \otimes I_n)\right)\right]. \tag{21}$$

Considering the relationship

$$\mathbb{E}_{\mathcal{U}\sim\mathbb{U}}\left[(U \otimes U)A(U^\dagger \otimes U^\dagger)\right] = \left(\frac{\text{Tr}(A)}{d^2-1} - \frac{\text{Tr}(\text{Swap}A)}{d(d^2-1)}\right) I_n \otimes I_n + \left(\frac{\text{Tr}(\text{Swap}A)}{d^2-1} - \frac{\text{Tr}(A)}{d(d^2-1)}\right)\text{Swap} \tag{22}$$

$$= \alpha I_n \otimes I_n + \beta\text{Swap}$$

where $\mathbb{U}$ is unitary 2-design, then Eq. 21 can be further calculated by

$$\mathbb{E}_{\mathcal{U}_1} \text{Tr}\,(\rho_{1,1}\rho_{2,1}) = \sum_{l=1}^{r} \mathbb{E}_{\mathcal{U}_1} \text{Tr}\left[\text{Swap}\left((K_l \otimes I_n)(\alpha I_n \otimes I_n + \beta\text{Swap})(K_l^\dagger \otimes I_n)\right)\right]$$

$$= \alpha \sum_{l=1}^{r} \text{Tr}\left[\text{Swap}(K_l K_l^\dagger \otimes I_n)\right] + \beta \sum_{l=1}^{r} \text{Tr}\left[\text{Swap}(K_l \otimes I_n)\text{Swap}(K_l^\dagger \otimes I_n)\right] \tag{23}$$

$$= \alpha \sum_{l=1}^{r} \text{Tr}\left[K_l K_l^\dagger\right] + \beta \sum_{l=1}^{r} |\text{Tr}\,[K_l]|^2.$$

The last equality comes from

$$\text{Tr}\left[\text{Swap}(K_l \otimes I_n)\text{Swap}(K_l^\dagger \otimes I_n)\right] = \text{Tr}\left[\text{Swap}(K_l \otimes I_n)(I_n \otimes K_l^\dagger)\right]$$

$$= \text{Tr}\left[\text{Swap}(K_l \otimes K_l^\dagger)\right] \tag{24}$$

$$= |\text{Tr}\,[K_l]|^2.$$

Consider $\sum_{l=1}^{r} \text{Tr}\left[K_l K_l^\dagger\right] = \text{Tr}[\mathcal{E}(I_n)] = d$ and denote $F = \sum_{l=1}^{r} |\text{Tr}\,[K_l]|^2$, Eq. 21 can be finally expressed as

$$\mathbb{E}_{\mathcal{U}_1} \text{Tr}\,(\rho_{1,1}\rho_{2,1}) = \left(\frac{1}{d^2-1} - \frac{\text{Tr}(\rho_1\rho_2)}{d(d^2-1)}\right)\text{Tr}[\mathcal{E}(I_n)] + \left(\frac{\text{Tr}(\rho_1\rho_2)}{d^2-1} - \frac{1}{d(d^2-1)}\right)F$$

$$= \frac{F-1}{d^2-1}\text{Tr}(\rho_1\rho_2) + \frac{1}{d^2-1}\left(d - F/d\right), \tag{25}$$

where $d = 2^n$. Therefore, we obtain a recursive formula for the overlap of the output states, as defined in 25. We have

$$\mathbb{E}_{\mathcal{U}_1,\mathcal{U}_1,...,\mathcal{U}_{\tilde{R}}} \text{Tr}\left(\rho_{1,\tilde{R}}\rho_{2,\tilde{R}}\right) = \frac{F-1}{d^2-1}\mathbb{E}_{\mathcal{U}_1,\mathcal{U}_1,...,\mathcal{U}_{\tilde{R}-1}} \text{Tr}(\rho_{1,\tilde{R}-1}\rho_{2,\tilde{R}-1}) + \frac{1}{d^2-1}\left(d - F/d\right). \tag{26}$$

Then, we can use this iteration relationship to construct a geometric sequence, that is

$$\mathbb{E}_{\mathcal{U}_1,\mathcal{U}_1,...,\mathcal{U}_{\tilde{R}}} \text{Tr}\left(\rho_{1,\tilde{R}}\rho_{2,\tilde{R}}\right) - \frac{1}{d} = \left(\frac{F-1}{d^2-1}\right)\left(\mathbb{E}_{\mathcal{U}_1,\mathcal{U}_1,...,\mathcal{U}_{\tilde{R}-1}} \text{Tr}(\rho_{1,\tilde{R}-1}\rho_{2,\tilde{R}-1}) - \frac{1}{d}\right)$$

$$= \left(\frac{F-1}{d^2-1}\right)^{\tilde{R}-1}\left(\mathbb{E}_{\mathcal{U}_1} \text{Tr}\,(\rho_{1,1}\rho_{2,1}) - \frac{1}{d}\right)$$

$$= \left(\frac{F-1}{d^2-1}\right)^{\tilde{R}-1}\left(\frac{F-1}{d^2-1} + \frac{d^2-F}{d(d^2-1)} - \frac{1}{d}\right) \tag{27}$$

$$= \left(\frac{F-1}{d^2-1}\right)^{\tilde{R}-1}\frac{F-1}{d(d+1)},$$

where the last equality comes from initial states $\rho_1 = \rho_2 = |0^n\rangle\langle0^n|$. Generally, $F \le d^2$, then if

$$\tilde{R} \le \frac{\log\left(\frac{F-1}{d(d+1)}(\eta - 1/d)^{-1}\right)}{\log(d^2-1) - \log(F-1)}, \tag{28}$$

we have $\mathbb{E}_{\mathcal{U}_1,\mathcal{U}_1,...,\mathcal{U}_{\tilde{R}}} \text{Tr}\left(\rho_{1,\tilde{R}}\rho_{2,\tilde{R}}\right) \ge \eta$.

# G PROOF OF THEOREM 1

Consider $U(\boldsymbol{\alpha}) = \prod_{r=1}^{LR} U(\boldsymbol{\alpha}_r)$ is composed of $LR$ two-qubit gates

$$U(\boldsymbol{\alpha}_r) = \exp\left(-i\sum_{j_1,j_2=0}^{4} \alpha_r(j_1,j_2)\left(P_{j_1} \otimes P_{j_2}\right)\right) = \exp\left(-i\langle\boldsymbol{\alpha}_r, \boldsymbol{P}_r\rangle\right), \tag{29}$$

where $P_j \in \{I, X, Y, Z\}$ and each $\alpha_r(j_1, j_2) \in [-1, 1]$ [8]. Using Taylor series, one obtains

$$U(\boldsymbol{\alpha}) = \prod_{r=1}^{LR}\sum_{k=0}^{\infty}\frac{(-i\langle\boldsymbol{\alpha}_r, \boldsymbol{P}_r\rangle)^k}{k!}. \tag{30}$$

Denote

$$U(\boldsymbol{\alpha}_r)_{\mathrm{tr}} = \sum_{k=0}^{K}\frac{(-i\langle\boldsymbol{\alpha}_r, \boldsymbol{P}_r\rangle)^k}{k!}, \tag{31}$$

therefore $U(\boldsymbol{\alpha}_r) - U(\boldsymbol{\alpha}_r)_{\mathrm{tr}} = \sum_{k=K+1}^{\infty}\frac{(-i\langle\boldsymbol{\alpha}_r, \boldsymbol{P}_r\rangle)^k}{k!}$. For arbitrary bit-string $x, y$, we can apply standard bound on Taylor series to bound

$$\|\langle x|(U(\boldsymbol{\alpha}_r) - U(\boldsymbol{\alpha}_r)_{\mathrm{tr}})|y\rangle\|_1 \le \kappa/K! \tag{32}$$

for some constant $\kappa$. Therefore we have

$$\langle 0^n|U^\dagger(\boldsymbol{\alpha})\rho U(\boldsymbol{\alpha})|0^n\rangle = \sum_{i,j=0}^{2^n-1}\rho_{ij}\langle 0^n|U^\dagger(\boldsymbol{\alpha})|i\rangle\langle j|U(\boldsymbol{\alpha})|0^n\rangle$$

$$= \sum_{i,j=0}^{2^n-1}\rho_{ij}\left(\sum_{\substack{y_1,y_2,\dots y_{LR-1}\in\{0,1\}^n \\ y_{LR}=i}}\prod_{r=1}^{LR}\langle 0^n|U(\boldsymbol{\alpha}_r)|y_r\rangle\right)\left(\sum_{\substack{y_1,y_2,\dots y_{LR-1}\in\{0,1\}^n \\ y_{LR}=j}}\prod_{r=1}^{LR}\langle y_r|U(\boldsymbol{\alpha}_r)|0^n\rangle\right)$$

$$= \sum_{i,j=0}^{2^n-1}\rho_{ij}\left(\sum_{\substack{y_1,y_2,\dots y_{LR-1}\in\{0,1\}^n \\ y_{LR}=i}}\prod_{r=1}^{LR}\langle 0^n|\sum_{k=0}^{\infty}\frac{(-i\langle\boldsymbol{\alpha}_r, \boldsymbol{P}_r\rangle)^k}{k!}|y_r\rangle\right)\cdot \tag{33}$$

$$\left(\sum_{\substack{y_1,y_2,\dots y_{LR-1}\in\{0,1\}^n \\ y_{LR}=j}}\prod_{r=1}^{LR}\langle y_r|\sum_{k=0}^{\infty}\frac{(-i\langle\boldsymbol{\alpha}_r, \boldsymbol{P}_r\rangle)^k}{k!}|0^n\rangle\right),$$

where the $y_1, y_2, \dots$ represent the Feymann integration path. According to inequality 32, $\langle y_r|U(\boldsymbol{\alpha}_r)|0^n\rangle$ can be approximated by a polynomial of degree $K$ based on Taylor truncated method, the above expression can be rewritten by

$$\sum_{r,s=0}^{2^n-1}\rho_{rs}\left(f_r(\boldsymbol{\alpha}_1,\dots\boldsymbol{\alpha}_{LR}) + \mathcal{O}\left(\frac{2^{LRn}}{(K!)^{LR}}\right)\right)\left(f_s(\boldsymbol{\alpha}_1,\dots\boldsymbol{\alpha}_{LR}) + \mathcal{O}\left(\frac{2^{LRn}}{(K!)^{LR}}\right)\right), \tag{34}$$

where $f_r(\boldsymbol{\alpha}_1,\dots\boldsymbol{\alpha}_{LR})$ represents a multi-variable polynomial of degree $LRK$.

Furthermore, we will show that Eq. 34 can be approximated by a low-degree function with at most $\binom{LR}{q}(K)^q$ terms, where $q = \mathcal{O}(1)$. To show this fact, we rewrite Eq. 34 by

$$\sum_{r,s=0}^{2^n-1}\rho_{rs}f_r(\boldsymbol{\alpha}_1,\dots\boldsymbol{\alpha}_{LR})f_s(\boldsymbol{\alpha}_1,\dots\boldsymbol{\alpha}_{LR}). \tag{35}$$

---

[8]Without loss of generality, we assume $\alpha_r(j_1, j_2) \in [-1, 1]$. For rotation parameters $|\alpha_r(j_1, j_2)| \in [1, 2\pi]$, we can repeat the related two-qubit gate constant times to keep all rotation angles fixing in the interval $[-1, 1]$.

For the convenience of the proof, we denote $f_r(\boldsymbol{\alpha}_1, ...\boldsymbol{\alpha}_{LR}) = f_r$. Let

$$f_r f_s = 4^{LRn} \sum_{\vec{i}} \sum_{j_1, j_2} \boldsymbol{a}_{\vec{i}} \alpha_1^{i_1}(j_1, j_2) \cdots \alpha_{LR}^{i_{LR}}(j_1, j_2),$$

where the factor $4^{LRn}$ aims to 'normalize' each term in the summation, $\vec{i} = (i_1, ..., i_{LR})$, $\vec{\boldsymbol{\alpha}} = (\alpha_1(j_1, j_2), ..., \alpha_{LR}(j_1, j_2))$, $j_1, j_2 \in [15]$ and each $0 \leq i_l \leq 2K$, $l \in [LR]$. For the convenience of the proof, we omit the index $(j_1, j_2)$ in $f_r f_s$ in the following procedure.

Given a constant value $q \leq \mathcal{O}(1)$, for every term like $\boldsymbol{a}_{\vec{i}} \alpha_1^{i_1} \cdots \alpha_v^{i_v}$ with $i_1 = \cdots = i_v \geq K$ and $v > q$, its corresponding parameter $|\boldsymbol{a}_{\vec{i}}| \leq 1/(K!)^q$ (based on Taylor series). Truncate above high-degree terms, and denote

$$\tilde{f}_{r,s} = \sum_{\substack{j_1, ..., j_q \leq K-1 \\ s_1 \leq s_2 ... \leq s_q \leq LR}} \boldsymbol{a}_{\vec{j}, \vec{s}} \alpha_{s_1}^{j_1} \cdots \alpha_{s_q}^{j_q}, \tag{36}$$

therefore, the relationship

$$\left| f_r f_s - \tilde{f}_{r,s} \right| = 4^{LRn} \left| \sum_{\vec{i}} \boldsymbol{a}_{\vec{i}} \alpha_1^{i_1} \cdots \alpha_{LR}^{i_{LR}} - \sum_{\substack{j_1, ..., j_q \leq K-1 \\ s_1 \leq s_2 ... \leq s_q \leq LR}} \boldsymbol{a}_{\vec{j}, \vec{s}} \alpha_{s_1}^{j_1} \cdots \alpha_{s_q}^{j_q} \right| \leq 4^{LRn} \left( \frac{K^{LR}}{(K!)^q} \right) \tag{37}$$

holds. Above inequality comes from the fact that there are $\mathcal{O}(K^{LR})$ terms in $f_r$ where the norm of each truncated term is upper bounded by $\leq \mathcal{O}(1/(K!)^q)$.

Then let $q = \mathcal{O}(1)$, $LR = \mathcal{O}(n \log(n))$, $\tilde{f}_{r,s}$ can provide an estimation to $f_r f_s$ within $2^{-\text{poly}(n)}$ additive error according to the Stirling's formula. Specifically, let $q = 1$ and use Stirling's formula, the error

$$4^{LRn} \left( \frac{K^{LR}}{(K!)^q} \right) = \frac{4^{n^2 \log n} K^{n \log n}}{K!} \approx \frac{4^{n^2 \log n} K^{n \log n}}{\sqrt{2\pi K}(K/e)^K}. \tag{38}$$

Let $K = n^2 \log n$, one obtains

$$\begin{aligned}
\frac{4^{n^2 \log n}(n^2 \log n)^{n \log n}}{\sqrt{2\pi n^2 \log n}(n^2 \log n/e)^{n^2 \log n}} &\leq \frac{(4^{\log n})^{n^2 \log n}(n^2 \log n)^{n \log n}}{(n^2 \log n/e)^{n^2 \log n}\sqrt{2\pi n^2 \log n}} \\
&= \frac{(n^2 \log n)^{n \log n}}{\sqrt{2\pi n^2 \log n}} \left( \frac{e}{\log n} \right)^{n^2 \log n} \\
&= \frac{1}{\sqrt{2\pi n^2 \log n}} \left( \frac{(e^n n^2 \log n)}{(\log n)^n} \right)^{n \log n} \\
&\leq \frac{1}{\sqrt{2\pi n^2}} 2^{-n^2},
\end{aligned} \tag{39}$$

where the last inequality holds for large $n$. Then Eq. 34 can be represented by a muti-variable polynomial function $f(\vec{\boldsymbol{\alpha}}, \rho)$ with $LR$ variables and at most $\text{poly}(n)$ terms, and the relationship

$$\left| \langle 0^n | U^\dagger(\vec{\boldsymbol{\alpha}}) \rho U(\vec{\boldsymbol{\alpha}}) | 0^n \rangle - f(\vec{\boldsymbol{\alpha}}, \rho) \right| \leq 2^{-n^2} \tag{40}$$

holds. Suppose the target observable $M = U(\boldsymbol{\alpha}^*)|0^n\rangle\langle 0^n|U^\dagger(\boldsymbol{\alpha}^*)$. Then we may write

$$\text{Tr}(M\rho) \approx f(\vec{\boldsymbol{\alpha}}^*, \rho) = \sum_{\substack{j_1, ..., j_q \leq (K-1) \\ s_1 \leq s_2 ... \leq s_q \leq LR}} \boldsymbol{b}_{\vec{j}, \vec{s}}(\rho) \alpha_{s_1}^{j_1, *} \cdots \alpha_{s_q}^{j_q, *}, \tag{41}$$

where $\boldsymbol{b}_{\vec{j}, \vec{s}}(\rho) = \sum_{s, r=0}^{2^n - 1} \rho_{r,s} \boldsymbol{a}_{\vec{j}, \vec{s}}$. On other hand, consider a machine learning procedure with data points $\{(x_i = \vec{\boldsymbol{\alpha}}_i, y_i = f(\vec{\boldsymbol{\alpha}}_i, \rho))\}$. Let the feature map $\Psi(\vec{\alpha}) = (\alpha_{s_1}^{j_1} \cdots \alpha_{s_q}^{j_q})_{\vec{s}, \vec{j}}$, and the target is

to synthesis the function $f(\vec{\boldsymbol{\alpha}}, \rho) = \langle \vec{\boldsymbol{b}}(\rho), \Psi(\vec{\boldsymbol{\alpha}}) \rangle$, where $\vec{\boldsymbol{b}}(\rho) \in \mathbb{R}^{LRn^2}$. Consider the loss function

$$\min_{\vec{\boldsymbol{b}}(\rho)} \lambda \langle \vec{\boldsymbol{b}}(\rho), \vec{\boldsymbol{b}}(\rho) \rangle + \sum_{i=1}^{N} \left( \langle \vec{\boldsymbol{b}}(\rho), \Psi(\vec{\boldsymbol{\alpha}}_i) \rangle - y_i \right)^2, \tag{42}$$

where $\lambda > 0$ is a hyper-parameter. Define the feature matrix $\Psi = (\Psi(\vec{\boldsymbol{\alpha}}_1), \dots, \Psi(\vec{\boldsymbol{\alpha}}_N))$ and the kernel matrix

$$\mathrm{K} = \Psi^\dagger \Psi = [\mathrm{K}(\vec{\boldsymbol{\alpha}}_i, \vec{\boldsymbol{\alpha}}_j)]_{i,j=1}^{N}, \tag{43}$$

where the kernel function

$$\mathrm{K}(\boldsymbol{\alpha}, \boldsymbol{\alpha}') = \sum_{l=0}^{K} \sum_{1 \le i_1 < \cdots < i_q \le LR} (\alpha_{i_1} \alpha'_{i_1} + \cdots + \alpha_{i_q} \alpha'_{i_q})^l. \tag{44}$$

Without loss of generality, we can normalize $\mathrm{K}(\boldsymbol{\alpha}, \boldsymbol{\alpha}')$ enabling $\mathrm{Tr}(\mathrm{K}) = \mathrm{N}$. Therefore, the optimal solution

$$\vec{\boldsymbol{b}}(\rho)_{\mathrm{opt}} = \sum_{i=1}^{N} \sum_{j=1}^{N} \Psi(\vec{\boldsymbol{\alpha}}_i) (\mathrm{K} + \lambda \mathrm{I})_{ij}^{-1} f(\vec{\boldsymbol{\alpha}}_j). \tag{45}$$

As a result, the trained machine learning model

$$\begin{aligned}
g(\vec{\boldsymbol{x}}) = \langle \vec{\boldsymbol{b}}(\rho)_{\mathrm{opt}}, \Psi(\vec{\boldsymbol{x}}) \rangle &= \sum_{i=1}^{N} \sum_{j=1}^{N} (\mathrm{K} + \lambda \mathrm{I})_{ij}^{-1} \mathrm{K}(\vec{\boldsymbol{\alpha}}_i, \vec{\boldsymbol{x}}) f(\vec{\boldsymbol{\alpha}}_j) \\
&= \sum_{j=1}^{N} \left( \sum_{i=1}^{N} (\mathrm{K} + \lambda I)_{ij}^{-1} \mathrm{K}(\vec{\boldsymbol{\alpha}}_i, \vec{\boldsymbol{x}}) \right) f(\vec{\boldsymbol{\alpha}}_j) \\
&= \sum_{j=1}^{N} \vec{\boldsymbol{\beta}}_j(\boldsymbol{x}) f(\vec{\boldsymbol{\alpha}}_j) \\
&= \sum_{j=1}^{N} \vec{\boldsymbol{\beta}}_j(\boldsymbol{x}) \langle 0^n | U^\dagger(\vec{\boldsymbol{\alpha}}_j) \rho U(\vec{\boldsymbol{\alpha}}_j) | 0^n \rangle + \mathcal{O}\left( \frac{2^{LRn}}{(K!)^{LR}} \right).
\end{aligned} \tag{46}$$

Now we analyze the prediction error of $g(\vec{\boldsymbol{x}})$ on the domain $[0, 2\pi]^{LR}$. Denote

$$\begin{aligned}
\tilde{\epsilon}(\vec{\boldsymbol{x}}) = \epsilon(\vec{\boldsymbol{x}}) + |\mathrm{Tr}(M(\vec{\boldsymbol{x}})\rho) - f(\vec{\boldsymbol{x}}, \rho)| &= |g(\vec{\boldsymbol{x}}) - f(\vec{\boldsymbol{x}}, \rho)| + |\mathrm{Tr}(M(\vec{\boldsymbol{x}})\rho)) - f(\vec{\boldsymbol{x}}, \rho)| \\
&\le |g(\vec{\boldsymbol{x}}) - f(\vec{\boldsymbol{x}}, \rho)| + 2^{-\mathrm{poly}(n)},
\end{aligned} \tag{47}$$

and the expected prediction error

$$\mathbb{E}_{\vec{\boldsymbol{x}}} [\epsilon(\vec{\boldsymbol{x}})] = \frac{1}{N} \sum_{i=1}^{N} \epsilon(\vec{\boldsymbol{x}}_i) + \left( \mathbb{E}_{\vec{\boldsymbol{x}}} [\epsilon(\vec{\boldsymbol{x}})] - \frac{1}{N} \sum_{i=1}^{N} \epsilon(\vec{\boldsymbol{x}}_i) \right). \tag{48}$$

Using the Cauchy-Schwartz inequality, the above first term can be upper bounded by

$$\frac{1}{N} \sum_{i=1}^{N} \epsilon(\vec{\boldsymbol{x}}_i) \le \sqrt{\frac{\lambda^2 \sum_{i=1}^{N} \sum_{j=1}^{N} (\mathrm{K} + \lambda I)_{ij}^{-1} y_i y_j}{N}}. \tag{49}$$

Therefore, if the matrix $\mathrm{K}$ is invertable and hyper-parameter $\lambda = 0$, the training error is zero. Without loss of generality, we set $\lambda = 1/\mathrm{poly}(n)$.

The generalized error can be characterized by the Rademacher complexity Mohri et al. (2018), that is

$$\mathbb{E}_{\vec{\boldsymbol{x}}} \epsilon(\vec{\boldsymbol{x}}) - \frac{1}{N} \sum_{i=1}^{N} \epsilon(\vec{\boldsymbol{x}}_i) \le \sqrt{\frac{\sup(\mathrm{K}(x,x)) \|\vec{\boldsymbol{a}}_{\mathrm{opt}}\|_\Psi}{N}} = \sqrt{\frac{\|\vec{\boldsymbol{b}}(\rho)_{\mathrm{opt}}\|_\Psi}{N}}, \tag{50}$$

where $\|\vec{b}(\rho)\|_\Psi = \langle \vec{b}(\rho)_{\mathrm{opt}}, \vec{b}(\rho)_{\mathrm{opt}}\rangle$. Ideally, $\vec{b}(\rho)_{\mathrm{opt}} = (b_{\vec{j},\vec{s}}(\rho))$ for $j_1, ..., j_q \leq K - 1$, $s_1 \leq s_2... \leq s_q \leq LR$, and $\vec{b}(\rho)$ is induced by the polynomial kernel function, therefore the 2-norm of the vector $(b_{\vec{j},\vec{s}}(\rho))_{\vec{j},\vec{s}}$ can be used to estimate $\|\vec{b}(\rho)\|_\Psi$. Noting that the multi-variable polynomial function $\sum_{\vec{j},\vec{s}} b_{\vec{j},\vec{s}}(\rho)\alpha_{s_1}^{j_1}\cdots\alpha_{s_q}^{j_q}$ belongs to $[0, 1]$ for all $\vec{\alpha} \in [0, 2\pi]^{LR}$. Let $\vec{\alpha}$ takes value from the bitstring $\{0, 1\}^{LR}$, we know that all $\left|b_{\vec{j},\vec{s}}(\rho)\right| \in [0, 1]$. Therefore, we can upper bound $\|\vec{b}(\rho)\|_\Psi$ by $(LR)K = (LR)n^2\log n$.

Combine all together, for any $M(\vec{x}) = U(\vec{x})|0^n\rangle\langle 0^n|U^\dagger(\vec{x})$ for $U \in \mathcal{U}_\mathcal{A}(R)$ and arbitrary density matrix $\rho$, we have the relationship

$$\mathbb{E}_{\vec{x}}\left|\sum_{j=1}^N \vec{\beta}_j(\boldsymbol{x})\langle 0^n|U^\dagger(\vec{\alpha}_j)\rho U(\vec{\alpha}_j)|0^n\rangle - \mathrm{Tr}\left(M(\vec{x})\rho\right)\right| \leq \sqrt{\frac{\lambda^2\sum_{i=1}^N\sum_{j=1}^N(\mathrm{K}+\lambda I)_{ij}^{-1}y_iy_j}{N}} + \sqrt{\frac{LRn^2\log n}{N}} \tag{51}$$

In the above first term, hyper-parameter $\lambda$ can take arbitrary value, and $\lambda = \sqrt{\lambda_{\min}(\mathrm{K})}/(nN)$ enables that the first term is upper bounded by $1/n$, where $\lambda_{\min}(\mathrm{K})$ represents the minimum eigenvalue of the kernel matrix K. In the second term, let $N = \tilde{\mathcal{O}}\left((LR)n^2\epsilon^{-2}\right)$, and the above error can be upper bounded by $\epsilon$.

# H PROOF OF QUANTUM LEARNING PRINCIPLES

## H.1 PROOF OF LEMMA 2

The proof of Lemma 2 depends on the intrinsic structure of the QCA model. Let $M_{\mathrm{opt}}$ represent an observable

$$M_{\mathrm{opt}} = \arg\max_{M=V|0^n\rangle\langle 0^n|V^\dagger, V\in\mathcal{U}_\mathcal{A}(R)}\left|\mathrm{Tr}\left(M(\rho_{p,\tilde{R}} - I_n/2^n)\right)\right|. \tag{52}$$

Given the QCA circuit set $\hat{\rho}_{\mathrm{QCA}}(R, \mathcal{A}, N) = \{U(\vec{\alpha}_j)\}_{j=1}^N$ with $N = LRn^2\epsilon^{-2}$, the intrinsic structure of QCA promises that there exists a vector $\vec{\beta}^{\mathrm{opt}}$ such that

$$\left|\sum_{j=1}^N \vec{\beta}_j^{\mathrm{opt}}(\vec{x})\mathrm{Tr}\left(U(\vec{\alpha}_j)P_0U^\dagger(\vec{\alpha}_j)\rho\right) - \mathrm{Tr}\left(M_{\mathrm{opt}}\rho\right)\right| \leq \epsilon \tag{53}$$

for any $n$-qubit density matrix $\rho$ and projector $P_0 = |0^n\rangle\langle 0^n|$. Denote

$$M_R(\vec{\beta}) = \sum_{i=1}^N \beta_i U(\vec{\alpha}_i)P_0U^\dagger(\vec{\alpha}_i),$$

according to the assumption in Lemma 2, we have

$$\begin{aligned}
\epsilon + \tilde{\epsilon} &< \min_{\vec{\beta}}\left|\mathbb{E}_{|\Psi_i\rangle\sim(\hat{\rho}_{\mathrm{QCA}},\vec{q})}\left[\mathrm{Tr}(M(\vec{\beta})(|\Psi_i\rangle\langle\Psi_i| - \rho_{p,\tilde{R}}))\right]\right| \\
&\leq \left|\mathbb{E}_{|\Psi_i\rangle\sim(\hat{\rho}_{\mathrm{QCA}},\vec{q})}\left[\mathrm{Tr}(M(\vec{\beta}_{\mathrm{opt}})(|\Psi_i\rangle\langle\Psi_i| - \rho_{p,\tilde{R}}))\right]\right| \\
&\leq \left|\mathbb{E}_{|\Psi_i\rangle\sim(\hat{\rho}_{\mathrm{QCA}},\vec{q})}\left[\mathrm{Tr}(M_{\mathrm{opt}}(|\Psi_i\rangle\langle\Psi_i| - \rho_{p,\tilde{R}}))\right] + \epsilon\right| \\
&= \left|\mathbb{E}_{|\Psi_i\rangle\sim(\hat{\rho}_{\mathrm{QCA}},\vec{q})}\left[\mathrm{Tr}(M_{\mathrm{opt}}(|\Psi_i\rangle\langle\Psi_i| - I_n/2^n))\right] - \mathrm{Tr}(M_{\mathrm{opt}}(\rho_{p,\tilde{R}} - I_n/2^n)) + \epsilon\right| \\
&\leq 1 - \frac{1}{2^n} - \mathrm{Tr}(M_{\mathrm{opt}}(\rho_{p,\tilde{R}} - I_n/2^n)) + \epsilon,
\end{aligned} \tag{54}$$

where the third line comes from the intrinsic structure of specific quantum circuit architecture, the last inequality comes from $\sum_i q_i = 1$ and $\langle\Psi_i|M_{\mathrm{opt}}|\Psi_i\rangle \leq 1$. As a result, we have

$$\mathrm{Tr}(M_{\mathrm{opt}}(\rho_{p,\tilde{R}} - I_n/2^n)) < 1 - \frac{1}{2^n} - \tilde{\epsilon}. \tag{55}$$

## H.2 PROOF OF LEMMA 1

According to the assumption, the relationship

$$
\begin{aligned}
\epsilon &\geq \max_{\vec{q}, M(\vec{\beta})} \left| \mathbb{E}_{|\Psi_i\rangle \sim (\hat{\rho}_{\text{QCA}}, \vec{q})} \text{Tr}\left(M(\vec{\beta})(|\Psi_i\rangle\langle\Psi_i| - \rho_{p,\tilde{R}})\right)\right| \\
&= \max_{\vec{q}, M(\vec{\beta})} \left| \mathbb{E}_{|\Psi_i\rangle \sim (\hat{\rho}_{\text{QCA}}, \vec{q})} \left[\text{Tr}\left(M(\vec{\beta})(|\Psi_i\rangle\langle\Psi_i| - \sigma)\right) - \text{Tr}\left(M(\vec{\beta})(\rho_{p,\tilde{R}} - \sigma)\right)\right]\right|
\end{aligned}
\tag{56}
$$

holds, where maximal entangled state $\sigma = I/d$, $d = 2^n$ and $M(\vec{\beta})$ is defined as Eq. 11. Randomly choose an index $t \in [N]$, and let

$$
M_t = \arg \max_{M = V|0\rangle\langle 0|V^\dagger} \text{Tr}\left(M(|\Psi_t\rangle\langle\Psi_t| - \sigma)\right),
\tag{57}
$$

where $V \in \mathcal{U}_\mathcal{A}(R)$. Based on the intrinsic structure in QCA, there exists a unit vector $\vec{\beta}^{(t)}$ such that

$$
\left|\text{Tr}\left(M_t(|\Psi_t\rangle\langle\Psi_t| - \sigma)\right) - \text{Tr}\left(M(\vec{\beta}^{(t)})(|\Psi_t\rangle\langle\Psi_t| - \sigma)\right)\right| \leq \epsilon.
\tag{58}
$$

Then assign the probability distribution $q_t = 1 - (N-1)/d$ and $q_j = 1/d$ for $j \neq t$. As a result, Eq. 56 can be further lower bounded by

$$
\left|\left(1 - \frac{N-1}{d}\right)\text{Tr}\left(M(\vec{\beta}^{(t)})(|\Psi_t\rangle\langle\Psi_t| - \sigma)\right) + \frac{1}{d}\sum_{j \neq t}\text{Tr}\left(M(\vec{\beta}^{(t)})(|\Psi_j\rangle\langle\Psi_j| - \sigma)\right) - \text{Tr}\left(M(\vec{\beta}^{(t)})(\rho_{p,\tilde{R}} - \sigma)\right)\right|.
\tag{59}
$$

Since $|\Psi_t\rangle$ is generated by a $R$-depth quantum circuit $U_t \in \mathcal{U}_\mathcal{A}(R)$, then $C_\epsilon(|\Psi_t\rangle) \leq LR$ which implies

$$
\text{Tr}\left[M_t\left(|\Psi_t\rangle\langle\Psi_t| - \frac{I}{d}\right)\right] \geq 1 - \frac{1}{d} - \epsilon,
\tag{60}
$$

then the relationship

$$
\text{Tr}\left[M(\vec{\beta}^{(t)})\left(|\Psi_t\rangle\langle\Psi_t| - \frac{I}{d}\right)\right] \geq \text{Tr}\left[M_t\left(|\Psi_t\rangle\langle\Psi_t| - \frac{I}{d}\right)\right] - \epsilon \geq 1 - \frac{1}{d} - 2\epsilon
\tag{61}
$$

holds. Therefore, $\left(1 - \frac{N-1}{d}\right)\text{Tr}\left(M(\vec{\beta}^{(t)})(|\Psi_t\rangle\langle\Psi_t| - \sigma)\right) \geq (1 - \frac{N-1}{d})(1 - \frac{1}{d} - 2\epsilon)$. Combining the result $\frac{1}{d}\sum_{j \neq t}\text{Tr}\left(M(\vec{\beta}^{(t)})(|\Psi_j\rangle\langle\Psi_j| - \sigma)\right) \geq \frac{-(N-1)}{d^2}$, where $\text{Tr}\left(M(\vec{\beta}^{(t)})(|\Psi_j\rangle\langle\Psi_j| - \sigma)\right) > -1/d$, we thus have

$$
\begin{aligned}
\text{Tr}\left(M(\vec{\beta}^{(t)})(\rho_{p,\tilde{R}} - \sigma)\right) &\geq \left(1 - \frac{N-1}{d}\right)\left(1 - \frac{1}{d} - 2\epsilon\right) + \frac{(N-1)}{d^2} - \epsilon \\
&= 1 - \frac{1}{d} - 3\epsilon - \frac{N-1}{d}\left(1 - \frac{2}{d} - 2\epsilon\right).
\end{aligned}
\tag{62}
$$

Note that $N = LRn^2 \log(n)\epsilon^{-2}$, $d = 2^n$, therefore

$$
\frac{N-1}{d}\left(1 - \frac{2}{d} - 2\epsilon\right) = \frac{\text{poly}(n)}{2^n}\left(1 - \frac{1}{2^n} - 2\epsilon\right) < \mathcal{O}(\epsilon)
\tag{63}
$$

for large $n \in \mathbb{Z}_{>0}$ and $\epsilon = 1/\text{poly}(n)$. Finally,

$$
\begin{aligned}
\max_{M = V|0^n\rangle\langle 0^n|V^\dagger} \left|\text{Tr}\left(M(\rho_{p,\tilde{R}} - I_n/2^n)\right)\right| &\geq \text{Tr}\left(M_t(\rho_{p,\tilde{R}} - I_n/2^n)\right) \\
&\geq \text{Tr}\left(M(\vec{\beta}^{(t)})(\rho_{p,\tilde{R}} - I_n/2^n)\right) - \epsilon \\
&\geq 1 - \frac{1}{d} - \mathcal{O}(\epsilon),
\end{aligned}
\tag{64}
$$

where the first inequality is valid since $M_t$ (defined by Eq. 57) is one of the instances in the set $\{V|0^n\rangle\langle 0^n|V^\dagger\}$, the second line comes from the intrinsic structure of QCA, and the third line comes from inequality 62. This implies $C_\epsilon^{\lim,\mathcal{A}}(\rho_{p,\tilde{R}}) \leq LR$.

## I   BAYESIAN OPTIMIZATION

### I.1   OPTIMIZATION SUBROUTINE

In the following, we show how to maximize the loss function $\mathcal{L}_R(\vec{\beta})$ via Bayesian optimization on a compact set. Bayesian optimization is composed by two significant components: $(i)$ a statistical model, in general *Gaussian process*, provides a posterior distribution conditioned on a prior distribution and a set of observations over $\mathcal{L}_R(\vec{\beta})$. $(ii)$ an *acquisition function* determines the position of the next sample point, based on the current posterior distribution over $\mathcal{L}_R(\vec{\beta})$.

Gaussian process is a set of random variables, where any subset forms a multivariate Gaussian distribution. For the optimization task considered in the main file, the random variables represent the value of the objective function $\mathcal{L}_R(\vec{\beta})$ at the point $\vec{\beta}$. As a distribution over $\mathcal{L}_R(\vec{\beta})$, a Gaussian process is completely specified by *mean function* and *covariance function*

$$
\begin{aligned}
\mu(\vec{\beta}) &= \mathbb{E}_{\vec{\beta}}[\mathcal{L}_R(\vec{\beta})] \\
k(\vec{\beta}, \vec{\beta}') &= \mathbb{E}_{\vec{\beta}}[(\mathcal{L}_R(\vec{\beta}) - \mu(\vec{\beta}))(\mathcal{L}_R(\vec{\beta}') - \mu(\vec{\beta}'))],
\end{aligned}
\tag{65}
$$

and the Gaussian process is denoted as $\mathcal{L}_R(\vec{\beta}) \sim \mathcal{GP}(\mu(\vec{\beta}), k(\vec{\beta}, \vec{\beta}'))$. Without loss of generality, we assume that the prior mean function $\mu(\vec{\beta}) = 0$. In the $t$-th iteration step, assuming observations $\mathrm{Acc}(t) = \{(\vec{\beta}^{(1)}, y(\vec{\beta}^{(1)})), \ldots, (\vec{\beta}^{(t)}, y(\vec{\beta}^{(t)}))\}$ are accumulated, where $y(\vec{\beta}^{(i)}) = \mathcal{L}_R(\vec{\beta}^{(i)}) + \epsilon_i$, with the quantum measurement error $\epsilon_i \sim \mathcal{N}(0, 1/4\hat{M})$ for $i \in [t]$, where $\hat{M}$ represents the measurement complexity in our algorithm. Conditioned on the accumulated observations $\mathrm{Acc}(t)$, the posterior distribution of $\mathcal{L}_R(\vec{\beta})$ is a Gaussian process with *mean function* $\mu_t(\vec{\beta}) = \mathbb{E}_{\vec{\beta}}[\mathcal{L}_R(\vec{\beta})|\mathrm{Acc}(t)]$ and *covariance function* $k_t(\vec{\beta}, \vec{\beta}') = \mathbb{E}_{\vec{\beta}}[(\mathcal{L}_R(\vec{\beta}) - \mu(\vec{\beta}))(\mathcal{L}_R(\vec{\beta}') - \mu(\vec{\beta}'))|\mathrm{Acc}(t)]$, specified by

$$
\begin{aligned}
\mu_t(\vec{\beta}) &= \boldsymbol{k}_t^{\mathsf{T}}[\boldsymbol{K}_t + \boldsymbol{I}_t/4\hat{M}]^{-1}\boldsymbol{y}_{1:t} \\
k_t(\vec{\beta}, \vec{\beta}') &= k(\vec{\beta}, \vec{\beta}') - \boldsymbol{k}_t^{\mathsf{T}}[\boldsymbol{K}_t + \boldsymbol{I}_t/4\hat{M}]^{-1}\boldsymbol{k}_t,
\end{aligned}
\tag{66}
$$

where $\boldsymbol{k}_t = [k(\vec{\beta}, \vec{\beta}^{(1)}) \quad \ldots \quad k(\vec{\beta}, \vec{\beta}^{(t)})]^{\mathsf{T}}$, the positive definite covariance matrix $\boldsymbol{K}_t = [k(\vec{\beta}, \vec{\beta}')]_{\vec{\beta}, \vec{\beta}' \in \vec{\beta}_{1:t}}$ with $\vec{\beta}_{1:t} = \{\vec{\beta}^{(1)}, \ldots, \vec{\beta}^{(t)}\}$ and $\boldsymbol{y}_{1:t} = [y(\vec{\beta}^{(1)}), \ldots, y(\vec{\beta}^{(t)})]^{\mathsf{T}}$. The posterior variance of $\mathcal{L}_R(\vec{\beta})$ is denoted as $\sigma_t^2(\vec{\beta}) = k_t(\vec{\beta}, \vec{\beta})$. The mean function $\mu_t(\vec{\beta})$ is related to the expected value of $\mathcal{L}_R(\vec{\beta})$, while the covariance $k_t$ estimates the deviations of $\mu_t(\vec{\beta})$ from the value of $\mathcal{L}_R(\vec{\beta})$. Then the prediction is obtained by conditioning the prior Gaussian process on the observations and returns a posterior distribution described by a Gaussian process multivariate distribution. Using the Sherman-Morrison-Woodbury formula (Seeger, 2004), the predictive distribution can be explicitly expressed as Eq. 66.

In the $t$-th iteration of Bayesian optimization, the acquisition function $\mathcal{A}(\vec{\beta})$ learns from the accumulated observations $\mathrm{Acc}(t-1)$ and leads the search to the next point $\vec{\beta}^{(t)}$ which is expected to gradually convergence to the optimal parameters of $\mathcal{L}_R(\vec{\beta})$. This procedure is achieved via maximizing $\mathcal{A}(\vec{\beta})$. In detail, the design of acquisition function should consider *exploration* (exploring domains where $\mathcal{L}_R(\vec{\beta})$ has high uncertainty) and *exploitation* (exploring domains where $\mathcal{L}_R(\vec{\beta})$ is expected to have large image value). The upper confidence bound is a widely used acquisition function, which is defined as

$$
\mathcal{A}_{\mathrm{UCB}}(\vec{\beta}) = \mu_{t-1}(\vec{\beta}) + \sqrt{\kappa_t}\sigma_{t-1}(\vec{\beta}),
\tag{67}
$$

and the next point $\vec{\beta}^{(t)}$ is decided by $\vec{\beta}^{(t)} = \arg\max_{\vec{\beta} \in \mathcal{D}_{\mathrm{domain}}} \mathcal{A}_{\mathrm{UCB}}(\vec{\beta})$. Here, $\kappa_t$ is a significant hyper-parameter, and a suitable $\kappa_t$ may lead $\vec{\beta}^{(t)}$ rapidly convergence to $\vec{\beta}_{opt}$. In Theorem 4, a specific $\kappa_t = 2N\log(t^2 N) + 2\log(t^2/\delta)$ is used. Details for maximizing $\mathcal{L}_R(\vec{\beta})$ are shown in Alg 2.

---

**Algorithm 2:** Bayesian Maximize Subroutine, $\mathrm{BMaxS}(\rho_{\mathrm{un}}, \hat{\rho}_{\mathrm{QCA}}(R, \mathcal{A}, N), T, \epsilon)$

---

**Input** : Noisy quantum state $\rho_{\mathrm{un}}$, a quantum state set $\hat{\rho}_{\mathrm{QCA}}(R, \mathcal{A}, N)$, failure probability
$\quad\quad\quad \delta \in (0, 1)$, iteration steps $T$, approximation error $\epsilon$
**Output:** True/False;

1 **Initialize** $\mu_0(\vec{\beta}) = 0$, $\sigma_0$, the covariance function $k(\cdot, \cdot)$;
2 **for** $t = 1, 2, ..., T$ **do**
3 $\quad$ Select $\kappa_t = 2N \log(t^2 N) + 2 \log(t^2/\delta)$;
4 $\quad$ Choose $\vec{\beta}^{(t)} = \arg \max_{\vec{\beta} \in \mathcal{D}_\beta} \mu_{t-1}(\vec{\beta}) + \sqrt{\kappa_t}\sigma_{t-1}(\vec{\beta})$;
5 $\quad$ Estimate $\mathcal{L}_R(\vec{\beta}^{(t)})$ with shadow tomography of $\rho_{\mathrm{un}}$ and $\hat{\rho}_{\mathrm{QCA}}$ (obtained from $\hat{M}$-snapshot
$\quad\quad$ measurements), that is $\left| y(\vec{\beta}^{(t)}) - \mathcal{L}_R(\vec{\beta}^{(t)}) \right| \leq \epsilon_t$ where $\epsilon_t \sim \mathcal{N}(0, 1/4\hat{M})$;
6 $\quad$ Update $\mu_t, \sigma_t^2$ as Eq. 66;
7 **if** $y(\vec{\beta}^{(T)}) \leq \epsilon$ **do**
8 $\quad$ **return** True
9 **else do**
10 $\quad$ **return** False

---

### I.2 PROOF OF THEOREM 4

**Theorem 5 (Formal version of Theorem 4)** *Take the weakly noisy state $\rho_{\mathrm{un}}$ and $\hat{\rho}_{\mathrm{QCA}}(R, \mathcal{A}, N)$ into Alg. 2. Pick the failure probability $\delta \in (0, 1)$ and let*

$$\kappa_t = 2N \log(t^2 N) + 2 \log(t^2/\delta) \tag{68}$$

*in the $t$-th iteration step, then the average regret $\mathrm{avr}_T$ can be upper bounded by*

$$\mathrm{avr}_T \leq \mathcal{O}\left( \sqrt{\frac{4N^2 \log^2 T + 2N \log T \log(\pi^2/(6\delta))}{T}} \right) \tag{69}$$

*with $1 - \delta$ success probability.*

We need following two lemmas to support our proof.

**Lemma 3 (Lemma 5.1 in Srinivas et al. (2012))** *Pick faliure probability $\delta \in (0, 1)$ and set $\kappa_t = 2 \log(|\mathcal{D}_{\mathrm{domain}}| \pi_t/\delta)$, where $\sum_{t \geq 1} \pi_t^{-1} = 1$ and $\pi_t > 0$. Then*

$$\left| \mathcal{L}(\vec{\beta}) - \mu_{t-1}(\vec{\beta}) \right| \leq \kappa_t^{1/2} \sigma_{t-1}(\vec{\beta}) \tag{70}$$

*holds for any $t \geq 1$ and $\vec{\beta} \in \mathcal{D}_{\mathrm{domain}}$.*

**Lemma 4 (Lemma 5.4 in Srinivas et al. (2012))** *Pick failure probability $\delta \in (0, 1)$ and let $\kappa_t$ be defined as in Lemma 3. Then the following holds with probability $\geq 1 - \delta$:*

$$\sum_{t=1}^{T} 4\kappa_t \sigma_{t-1}^2(\vec{\beta}_t) \leq \kappa_T \gamma_T, \tag{71}$$

*where $\gamma_T = \max_{A \in \mathcal{D}_{\mathrm{domain}}} \frac{1}{2} \log \left| I + \sigma^{-2} K_A \right|$, and $K_A$ represents the used covariance matrix in Bayesian optimization.*

We first consider the continuity of the loss function $\mathcal{L}(\vec{\beta})$. Considering $\sum_{i=1}^{N} \beta_i = 1$, the gradient function can be upper bounded by

$$\left| \frac{\partial \mathcal{L}(\vec{\beta})}{\partial \beta_j} \right| = \left| \mathbb{E}_{\hat{\rho}_i \sim \vec{q}} \mathrm{Tr} \left[ \hat{\rho}_j (\hat{\rho}_i - \rho_{\mathrm{un}}) \right] \right| \leq 1, \tag{72}$$

where the inequality comes from $|\text{Tr}\left[\hat{\rho}_j(\hat{\rho}_i - \rho_{\text{un}})\right]| \leq 1$. Therefore, the relationship

$$\left|\mathcal{L}(\vec{\boldsymbol{\beta}}) - \mathcal{L}(\vec{\boldsymbol{\beta}}')\right| \leq \left|\vec{\boldsymbol{\beta}} - \vec{\boldsymbol{\beta}}'\right|_1 \tag{73}$$

holds for any $\vec{\boldsymbol{\beta}}, \vec{\boldsymbol{\beta}}' \in \mathcal{D}_{\text{domain}}$. Now let us choose a discretization $\mathcal{D}_{\text{domain}}^t$ of size $(\tau_t)^N$ such that for all $\vec{\boldsymbol{\beta}} \in \mathcal{D}_{\text{domain}}$,

$$\|\vec{\boldsymbol{\beta}} - [\vec{\boldsymbol{\beta}}]_t\|_1 \leq N/\tau_t, \tag{74}$$

where $[\vec{\boldsymbol{\beta}}]_t$ represents the closest point in discretization on $\mathcal{D}_{\text{domain}}^t$ to $\vec{\boldsymbol{\beta}}$. Combine Eqs. 73-74, one obtains

$$\left|\mathcal{L}(\vec{\boldsymbol{\beta}}) - \mathcal{L}([\vec{\boldsymbol{\beta}}]_t)\right| \leq N\tau_t^{-1} = t^{-2}, \tag{75}$$

where the last inequality comes from $\tau_t = t^2 N$, furthermore, $|\mathcal{D}_{\text{domain}}^t| = (t^2 N)^N$.

Using Lemma 3, we know that

$$\kappa_t = 2\log(|\mathcal{D}_{\text{domain}}| a_t/\delta) \tag{76}$$

enables the relationship

$$\left|\mathcal{L}(\vec{\boldsymbol{\beta}}) - \mu_{t-1}(\vec{\boldsymbol{\beta}})\right| \leq \sqrt{\kappa_t}\sigma_{t-1}(\vec{\boldsymbol{\beta}}) \tag{77}$$

holds with probability at least $1 - \delta$, where $a_t > 0$ and $\sum_{t \geq 1} a_t^{-1} = 1$. A common selection is $a_t = \pi t^2/6$. Taking $|\mathcal{D}_{\text{domain}}^t| = (t^2 N)^N$ into $\kappa_t$ (Eq. 76), one obtains

$$\kappa_t = 2\log\left((t^2 N)^N a_t/\delta\right). \tag{78}$$

Combine Eq. 75 and 77, we have

$$\left|\mathcal{L}(\vec{\boldsymbol{\beta}}^*) - \mu_{t-1}([\vec{\boldsymbol{\beta}}^*]_t)\right| \leq \left|\mathcal{L}(\vec{\boldsymbol{\beta}}^*) - \mathcal{L}([\vec{\boldsymbol{\beta}}^*]_t)\right| + \left|\mathcal{L}([\vec{\boldsymbol{\beta}}^*]_t) - \mu_{t-1}([\vec{\boldsymbol{\beta}}^*]_t)\right| \leq t^{-2} + \sqrt{\kappa_t}\sigma_{t-1}([\vec{\boldsymbol{\beta}}^*]_t) \tag{79}$$

for any $t \geq 1$, where $\vec{\boldsymbol{\beta}}^* = \arg\max_{\vec{\boldsymbol{\beta}} \in \mathcal{D}_\beta} \mathcal{L}(\vec{\boldsymbol{\beta}})$. Now we connect the relationship between above inequality to the regret bound.

By the definition of $\vec{\boldsymbol{\beta}}^{(t)}$ (maximizing the $\mathcal{A}_{\text{UCB}}(\vec{\boldsymbol{\beta}})$ in the $t$-th step): $\mu_{t-1}(\vec{\boldsymbol{\beta}}^{(t)}) + \sqrt{\kappa_t}\sigma_{t-1}(\vec{\boldsymbol{\beta}}^{(t)}) \geq \mu_{t-1}([\vec{\boldsymbol{\beta}}^*]_t) + \sqrt{\kappa_t}\sigma_{t-1}([\vec{\boldsymbol{\beta}}^*]_t)$. Also, by Eq. 79, we have $\mathcal{L}(\vec{\boldsymbol{\beta}}^*) \leq \mu_{t-1}([\vec{\boldsymbol{\beta}}^*]_t) + \sqrt{\kappa_t}\sigma_{t-1}([\vec{\boldsymbol{\beta}}^*]_t) + 1/t^2$. Therefore, instantaneous regret

$$\begin{aligned}
r_t &= \mathcal{L}(\vec{\boldsymbol{\beta}}^*) - \mathcal{L}(\vec{\boldsymbol{\beta}}^{(t)}) \\
&\leq \sqrt{\kappa_t}\sigma_{t-1}(\vec{\boldsymbol{\beta}}^{(t)}) + 1/t^2 + \mu_{t-1}(\vec{\boldsymbol{\beta}}^{(t)}) - \mathcal{L}(\vec{\boldsymbol{\beta}}^{(t)}) \\
&\leq 2\sqrt{\kappa_t}\sigma_{t-1}(\vec{\boldsymbol{\beta}}^{(t)}) + 1/t^2,
\end{aligned}$$

where the last inequality comes from Eq. 77. Using Lemma 4, $\sum_{t=1}^T 4\kappa_t\sigma_{t-1}(\vec{\boldsymbol{\beta}}_t) \leq \kappa_T\gamma_T$, where

$$\gamma_T = \max_{A \in \mathcal{D}_{\text{domain}}} \frac{1}{2}\log\left|I + 2\hat{M}K_A\right|,$$

$K_A$ represents the used covariance matrix in $\text{BMaxS}(\rho_{p,\tilde{R}}, \hat{\rho}_{\text{QCA}}(R, \mathcal{A}, N), T)$ and $\hat{M}$ represents the number of measurement in generating the shadow tomography. In the linear function case, $K_A$ can be selected as the polynomial kernel function, and $\gamma_T = \mathcal{O}(N\log(T))$ Vakili et al. (2021). Furthermore, using the Cauchy-Schwartz inequality to $\sum_{t=1}^T 4\kappa_t\sigma_{t-1}^2(\vec{\boldsymbol{\beta}}_t) \leq \kappa_T\gamma_T$, one obtains $\sum_{t=1}^T 2\kappa_t^{1/2}\sigma_{t-1}(\vec{\boldsymbol{\beta}}_t) \leq \sqrt{T\kappa_T\gamma_T}$

Finally, we have the result

$$
\begin{aligned}
\mathrm{avr}_T &= \frac{1}{T}\sum_{t=1}^{T} r_t \\
&\leq \frac{1}{T}\sum_{t=1}^{T} 2\kappa_t^{1/2}\sigma_{t-1}(\vec{\beta}_t) + \frac{\pi^2}{6T} \\
&\leq \sqrt{\frac{\kappa_T \gamma_T}{T}} + \frac{\pi^2}{6T} \\
&= \mathcal{O}\left(\sqrt{\frac{2N^2 \log(T^2 N)\log(T) + 2N\log(T^2\pi^2/(6\delta))\log(T)}{T}}\right) \\
&\leq \mathcal{O}\left(\sqrt{\frac{4N^2 \log^2 T + 2N\log T \log(\pi^2/(6\delta))}{T}}\right),
\end{aligned} \tag{80}
$$

where the first inequality results from $\sum_{t\geq 1} t^{-2} = \pi^2/6$ and the last inequality comes from $N \leq T$.

## J    PROOF OF SAMPLE COMPLEXITY LOWER BOUND

we require the following lemmas to support our proof.

**Lemma 5 (Lemma 6 in Wang et al. (2021))** *Consider a single instanoise channel $\mathcal{N} = \mathcal{N}_1 \otimes \cdots \otimes \mathcal{N}_n$ where each local noise channel $\{\mathcal{N}_j\}_{j=1}^n$ is a Pauli noise channel that satisfies $\mathcal{N}_j(\sigma) = q_\sigma \sigma$ for $\sigma \in \{X, Y, Z\}$ and $q_\sigma$ be the Pauli strength. Then we have*

$$
D_2\left(\mathcal{N}(\rho)\|\frac{I^{\otimes n}}{2^n}\right) \leq q^{2c} D_2\left(\rho\|\frac{I^{\otimes n}}{2^n}\right), \tag{81}
$$

*where $D_2(\cdot\|\cdot)$ represents the 2-Renyi relative entropy, $q = \max_\sigma q_\sigma$ and $c = 1/(2\ln 2)$.*

**Lemma 6** *Given an arbitrary $n$-qubit density matrix and maximally mixed state $I^{\otimes n}/2^n$, we have*

$$
D\left(\rho\|I^{\otimes n}/2^n\right) \leq D_2\left(\rho\|I^{\otimes n}/2^n\right), \tag{82}
$$

*where $D(\cdot\|\cdot)$ denotes the relative entropy and $D_2(\cdot\|\cdot)$ denotes the 2-Renyi relative entropy.*

*Proof:* Given quantum states $\rho$ and $\sigma$, the quantum 2-Renyi entropy

$$
D_2(\rho\|\sigma) = \log \mathrm{Tr}\left[\left(\sigma^{-1/4}\rho\sigma^{-1/4}\right)^2\right]. \tag{83}
$$

When $\sigma = I^{\otimes n}/2^n$, we have $D_2(\rho\|I^{\otimes n}/2^n) = \log \mathrm{Tr}\left[\left((I^{\otimes n}/2^n)^{-1}\rho^2\right)\right] = n + \log \mathrm{Tr}[\rho^2]$. Noting that the function $y = x^2 - x\log x \geq 0$ when $x \in [0,1]$, and this implies $\mathrm{Tr}(\rho^2) \geq \mathrm{Tr}(\rho\log\rho)$. Finally, we have

$$
D\left(\rho\|I^{\otimes n}/2^n\right) = n + \mathrm{Tr}\left[\rho\log\rho\right] + n \leq \mathrm{Tr}\left[\rho^2\right] + n = D_2\left(\rho\|I^{\otimes n}/2^n\right). \tag{84}
$$

**Task 2** *Consider a pure quantum state $\rho_0$ and two quantum circuit $\mathcal{C}_1$ and $\mathcal{C}_2$ with $R_1$ and $R_2$ depth ($R_1 < R_2$), respectively. Each quantum circuit is affected by by $p$-strength Pauli channel in each layer. Suppose that a distinguisher is given access to copies of the quantum states $\Phi_{\mathcal{C}_1}(\rho_0)$ and $\Phi_{\mathcal{C}_2}(\rho_0)$, then what is the fewest number of copies sufficing to identify quantum states $\Phi_{\mathcal{C}_1}(\rho_0)$ and $\Phi_{\mathcal{C}_2}(\rho_0)$ with high probability?*

Obviously, if a quantum state complexity prediction problem can predict the complexity $R_1$ and $R_2$, then we can classify quantum states $\Phi_{\mathcal{C}_1}(\rho_0)$ and $\Phi_{\mathcal{C}_2}(\rho_0)$ easily. The sample complexity of Task 2

thus can be used to benchmark the sample complexity lower bound of the quantum state complexity prediction problem.

Now we prove the sample complexity lower bound to Task 2. We consider the sample complexity $m$ in distinguishing quantum states $\Phi_{\mathcal{C}_1}(\rho_0)$ and $\Phi_{\mathcal{C}_2}(\rho_0)$. When their trace distance is quite large, let $\delta \in (0, 1)$ and we have

$$
\begin{aligned}
1 - \delta &\le \frac{1}{2} \left\| \Phi_{\mathcal{C}_1}(\rho_0)^{\otimes m} - \Phi_{\mathcal{C}_2}(\rho_0)^{\otimes m} \right\|_1 \\
&\le \frac{1}{2} \left( \left\| \Phi_{\mathcal{C}_1}(\rho_0)^{\otimes m} - (I_n/2^n)^{\otimes m} \right\|_1 + \left\| \Phi_{\mathcal{C}_2}(\rho_0)^{\otimes m} - (I_n/2^n)^{\otimes m} \right\|_1 \right) \\
&\le \frac{1}{\sqrt{2}} \left( D^{1/2} \left( \Phi_{\mathcal{C}_1}^{\otimes m}(\rho_0) \| (I_n/2^n)^{\otimes m} \right) + D^{1/2} \left( \Phi_{\mathcal{C}_2}^{\otimes m}(\rho_0) \| (I_n/2^n)^{\otimes m} \right) \right),
\end{aligned}
\tag{85}
$$

where the second line comes from the triangle inequality and the third line comes from the Pinsker's inequality. Using Lemmas 5 and 6, we have

$$
\begin{aligned}
1 - \delta &\le \frac{1}{\sqrt{2}} \left( D_2^{1/2} \left( \Phi_{\mathcal{C}_1}^{\otimes m}(\rho_0) \| (I_n/2^n)^{\otimes m} \right) + D_2^{1/2} \left( \Phi_{\mathcal{C}_2}^{\otimes m}(\rho_0) \| (I_n/2^n)^{\otimes m} \right) \right) \\
&\le \frac{\sqrt{nm}}{\sqrt{2}} ((1-p)^{cR_1} + (1-p)^{cR_2}) \\
&\le \sqrt{2nm}(1-p)^{cR_1},
\end{aligned}
\tag{86}
$$

As a result we have

$$
m \ge \frac{(1-p)^{-2cR_1}(1-\delta)^2}{2n}.
\tag{87}
$$

## K   REVIEW OF SHADOW TOMOGRAPHY

In quantum computation, the basic operators are the Pauli operators $\{I, \sigma^x, \sigma^y, \sigma^z\}$ which provide a basis for the density operators of a single qubit as well as for the unitaries that can be applied to them. For an $n$-qubit case, one can construct the Pauli group according to

$$
\mathbf{P}_n = \{e^{i\alpha\pi/2} \sigma^{j_1} \otimes \dots \otimes \sigma^{j_n} | j_k \in \{I, x, y, z\}, 1 \le k \le n\},
$$

where $e^{i\alpha\pi/2} \in \{0, \pm 1, \pm i\}$ is a global phase. Then the Clifford group $\mathbf{Cl}(2^n)$ is defined as the group of unitaries that normalize the Pauli group:

$$
\mathbf{Cl}(2^n) = \left\{ U | U\mathbf{P}^{(1)}U^\dagger = \mathbf{P}^{(2)}, \left( \mathbf{P}^{(1)}, \mathbf{P}^{(2)} \in \mathbf{P}_n \right) \right\}.
$$

The $n$-qubit Clifford gates are then defined as elements in the Clifford group $\mathbf{Cl}(2^n)$, and these Clifford gates compose the Clifford circuit.

Randomly sampling Clifford circuit $U$ can reproduce the first 3 moments of the full Clifford group endowed with the Haar measure $\mathbf{d}\mu(U)$ which is the unique left- and right- invariant measure such that

$$
\int_{\mathbf{Cl}(2^n)} \mathbf{d}\mu(U) f(U) = \int \mathbf{d}\mu(U) f(VU) = \int \mathbf{d}\mu(U) f(UV)
$$

for any $f(U)$ and $V \in \mathbf{Cl}(2^n)$. Using this property, one can sample Clifford circuits $U \in \mathbf{Cl}(2^n)$ with the probability $\Pr(U)$, and the corresponding expectation $\mathrm{E}_{U \in \mathbf{Cl}(2^n)}[(U\rho U^\dagger)^{\otimes t}]$ can be expressed as

$$
\sum_{U \in \mathbf{Cl}(2^n)} \Pr(U) \left( U\rho U^\dagger \right)^{\otimes t} = \int_{\mathbf{Cl}(2^n)} \mathbf{d}\mu(U) \left( U\rho U^\dagger \right)^{\otimes t},
$$

for any $n$-qubit density matrix $\rho$ and $t = 1, 2, 3$, where $\Pr(U)$ indicates the probability on sampling $U$. The right hand side of the above equation can be evaluated explicitly by representation theory, this thus yields a closed-form expression for sampling from a Clifford group.

To extract meaningful information from an $n$-qubit unknown quantum state $\rho$, the shadow tomography technique was proposed by Huang et al. (2020). The Clifford sampling is implemented by

repeatedly performing a simple measurement procedure: apply a random unitary $U \in \mathbf{Cl}(2^n)$ to rotate the state $\rho$ and perform a $\sigma^z$-basis measurement. The number of repeating times of this procedure is defined as the Clifford sampling complexity. On receiving the $n$-bit measurement outcome $|b\rangle \in \{0,1\}^n$, according to the Gottesman-Knill theorem (Gottesman, 1997), we can efficiently store an classical description of $U^\dagger |b\rangle\langle b| U$ in the classical memory. This classical description encodes meaningful information of the state $\rho$ from a particular angle, and it is thus instructive to view the average mapping from $\rho$ to its classical snapshot $U^\dagger |b\rangle\langle b| U$ as a quantum channel:

$$\mathcal{M}(\rho) = \mathrm{E}_{U \in \mathbf{Cl}(2^n)} \left( \mathrm{E}_{b \in \{0,1\}^n} [U^\dagger |b\rangle\langle b| U] \right), \tag{88}$$

where the quantum channel $\mathcal{M}$ depends on the ensemble of unitary transformation, and the quantum channel $\mathcal{M}$ can be further expressed as

$$\mathcal{M}(\rho) = \mathrm{E}_{U \in \mathbf{Cl}(2^n)} \sum_{\widehat{b} \in \{0,1\}^n} \langle \widehat{b}| U \rho U^\dagger |\widehat{b}\rangle U^\dagger |\widehat{b}\rangle\langle \widehat{b}| U = \frac{\rho + \mathrm{Tr}(\rho) I}{(2^n + 1) 2^n}, \tag{89}$$

where $I$ indicates the $2^n \times 2^n$ identity matrix. Therefore the inverse of quantum channel $\mathcal{M}^{-1}(\rho) = (2^n + 1)\rho - I$, and an estimation of $\rho$ is defined as

$$\widehat{\rho} = \mathcal{M}^{-1} \left( U^\dagger |b\rangle\langle b| U \right).$$

Repeat this procedure $M$ times results in an array of Clifford samples of $\rho$:

$$S(\rho; M) = \left\{ \widehat{\rho}_1 = \mathcal{M}^{-1} \left( U_1^\dagger |b_1\rangle\langle b_1| U_1 \right), ..., \widehat{\rho}_M = \mathcal{M}^{-1} \left( U_M^\dagger |b_M\rangle\langle b_M| U_M \right) \right\}, \tag{90}$$

and $\frac{1}{M} \sum_{m=1}^{M} \hat{\rho}_m$ is defined as the *classical shadow* of the quantum state $\rho$.

## L   NUMERICAL SIMULATIONS FOR STRESS-TESTING

Here, we conducted stress testing of the quantum learning algorithm in this study. Specifically, we applied the algorithm to estimate the complexity lower bound of quantum states under noisy environments for one-dimensional dynamical evolution. We tested the transverse-field Ising model with system sizes of $1 \times 6$, $1 \times 8$, and $1 \times 10$, and the numerical results are presented in Figure 3. Our findings indicate that the complexity lower bound of noisy quantum circuits does not grow linearly with circuit depth, which is consistent to the results given by Figure 2 in the main file.

Furthermore, we observed that as the system size and complexity increase, the curves corresponding to two different quantum states, $\tilde{R} = 2$ and $\tilde{R} = 10$, exhibit similar overall trends for the system size $n \in \{6, 8, 10\}$. Moreover, the variance of the algorithm does not increase with the growing system size but remains consistently stable. At the same time, the gap between the curves of quantum states with different complexities does not show a closing trend as the system size grows. These characteristics demonstrate the robustness of our learning algorithm against increases in system size and complexity. Therefore, our algorithm remains reliable for predicting the complexity of large-scale weakly noisy quantum states.

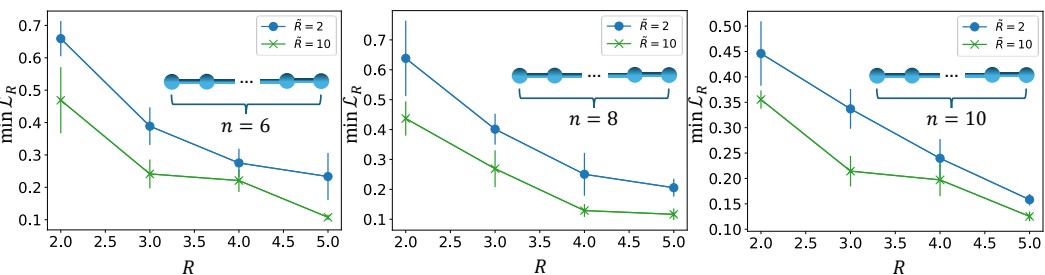

Figure 3: These three subfigures illustrate the trend of the function $\min_{\vec{\beta}} \mathcal{L}_R$ as it varies with the circuit depth $R$ of QCA set.In the stress-testing experiments, we focus on the Hamiltonian dynamics of 1D transverse field Ising models, with system sizes ranging from 6 to 10. In all cases, we observe that the weakly noisy state with $\tilde{R} = 2$ exhibits a larger quantum state complexity lower bound compared to the weakly noisy state with $\tilde{R} = 10$.

