# OpenReview forum: "Learning the Complexity of Weakly Noisy Quantum States"
_ICLR.cc/2025/Conference — ICLR 2025 Poster_

### Official Review · Reviewer_wYLf · 2024-10-29

**Soundness:** 3
**Presentation:** 2
**Contribution:** 2
**Rating:** 6
**Confidence:** 2

**Summary:**

The paper proposes a quantum learning algorithm that estimates the complexity of weakly noisy quantum states, unlike prior research focusing primarily on pure states, thereby connecting theoretical principles to practical scenarios observed in real-world quantum devices. The proposed algorithm is based on quantum circuit architecture (QCA) combined with classical shadow representation. It reaches near-optimal sample efficiency in the estimation of state complexity, which is theoretically analyzed in this paper to underscore its efficacy. The paper points out several promising research studies that may open up when the expressivity of QCA can be improved, and methodologies for direct complexity predictions in noisy quantum states are developed.

**Strengths:**

1. The paper confirms that the proposed approach can handle such complex quantum states even in noisy environments, validated through a robust theoretical framework based on theorems and proofs, including the sample complexity and the efficacy of Bayesian optimization in the noisy quantum environment.
2. The application of classical shadow representation in quantifying the complexity of weakly noisy quantum states is novel and crucial from a quantum perspective. This subsequently gives way to efficient quantum information processing classically, making the theoretical methods introduced more feasible.

**Weaknesses:**

1. The paper acknowledges that the proposed algorithm approximates the complexity of noisy quantum states and refers to further research on more direct prediction methods. However, it would benefit from an in-depth analysis of limitations, such as sensitivity on different noise models or how scaling issues may affect deployment to larger quantum systems.
2. The paper lacks a detailed discussion of the specific hardware requirements and the practical steps needed for real-world implementation. Details such as compatibility with existing quantum hardware, scalability in practical quantum systems, and specific technological constraints are not thoroughly explored.

**Questions:**

1. Could the authors elaborate on specific quantum hardware requirements for implementing the algorithm? Are there particular quantum systems where the algorithm's performance has been or can be practically tested?

2. How does the algorithm perform under different types of quantum noise models? Is the accuracy in the complexity predictions different between, e.g., local-depolarizing versus global-depolarizing channels?

The paper could be enhanced by considering the following suggestions:

- The paper should include a more detailed discussion of the computational and practical limitations of the algorithm (for example, stress-testing the algorithm under increased complexities and reporting on its degradation or stability performance.)
- Incorporating key proofs and insights from the appendices into the main body would enhance the paper's value, as some essential concepts, like the subroutine BMaxS, are only briefly mentioned in the main text.
- It would also be very useful if the manuscript provided a framework for empirical testing and validation of the algorithm, including benchmark selection and metrics of validation. This would help bridge the missing gap between theoretical research and practical applications and give a much stronger justification for the effectiveness of the algorithm in noisy quantum environments.

---

> ### Author Response · Authors · 2024-11-20
> **Addressing "Questions/Weaknesses in Official Review of 3076 by  Reviewer wYLf" (part 1)**
>
> **Comment 1**: *Could the authors elaborate on specific quantum hardware requirements for implementing the algorithm? Are there particular quantum systems where the algorithm's performance has been or can be practically tested?*
>
> **Response:**  We thank the reviewer for this comment. We note that our learning algorithm has very few limitations to the practical quantum hardware. In the Alg.~1, we only require (i) the target weakly noisy quantum state $\rho_{\rm un}$, and (ii) the QCA state set $\hat{\rho}_{\rm QCA}$, which is essentially the classical shadow representation [Hsin-Yuan Huang et al., 2020]. When preparing the classical shadow, recent work has successfully integrated quantum error mitigation methods into the classical shadow protocol, providing rigorous theoretical guarantees in preparing shadows even in noisy environments [Hamza Jnane et al., 2024]. As a result, our Alg 1 only requires a quantum computer which supports the quantum error mitigation function, with approximately $50-100$ noisy qubits and $\geq 0.99$ quantum gate fidelity. Some popular quantum computation platforms may satisfy these requirements and can be used to prepare the required classical shadow, such as Eagle used in Ref. [Youngseok Kim et al., 2023] and Sycamore [Frank Arute et al., 2019]. This work is essentially a *theoretical research*, and the algorithm has not yet been explicitly tested on these practical platforms. In the future work, we are happy to explore how the algorithm performs on practical quantum hardware, which remains a key direction for further validation.
>
> References:
> Hsin-Yuan Huang, Richard Kueng, and John Preskill. Predicting many properties of a quantum system
> from very few measurements. Nature Physics, 16(10):1050–1057, 2020.
>
> Hamza Jnane, Jonathan Steinberg, Zhenyu Cai, H Chau Nguyen, and B´alint Koczor. Quantum error
> mitigated classical shadows. PRX Quantum, 5(1):010324, 2024.
>
> Youngseok Kim, Andrew Eddins, Sajant Anand, KenXuan Wei, Ewout van den Berg, Sami Rosenblatt, Hasan Nayfeh, Yantao Wu, Michael Zaletel, Kristan Temme, and Abhinav Kandala. Evidence for the utility of quantum computing before fault tolerance. Nature, 618:500–505, 2023.
>
> Frank Arute, Kunal Arya, Ryan Babbush, Dave Bacon, Joseph C Bardin, Rami Barends, Rupak Biswas, Sergio Boixo, Fernando GSL Brandao, David A Buell, et al. Quantum supremacy using a programmable superconducting processor. Nature, 574(7779):505–510, 2019.
>
> **Comment 2**: *How does the algorithm perform under different types of quantum noise models? Is the accuracy in the complexity predictions different between, e.g., local-depolarizing versus global-depolarizing channels?*
>
> **Response**:  We thank the reviewer for this interesting question. From a theoretical perspective, our algorithm is broadly applicable to a wide range of noise models where the local noise channel can be decomposed using a set of Kraus operators, and the noise strength is approximately $p=\mathcal{O}(n^{-1})$ as stated in Fact 1. Moreover, the theoretical guarantees provided by Theorems 1 and 2 impose no additional constraints on the noise model. We also note that the global depolarizing channel is essentially a special case of contracting $n$ local noise models. Specifically, given the global depolarizing channel $\mathcal{E}$ with noisy strength $p_g$ and local depolarizing channel
>
> $\otimes_j\mathcal{E}_j$ with strength $p_l$,
>
> according to the relative entropy inequality, for any input state $\rho$, we can observe the relationships
> \begin{align}
>     D(\mathcal{E}(\rho)\|I_n/2^n)\leq (1-p_g)D(\rho\|I_n/2^n),
> \end{align}
> and
> \begin{align}
>     D(\otimes_{j=1}^n\mathcal{E}_j(\rho)\|I_n/2^n)\leq (1-p_l)^nD(\rho\|I_n/2^n)
> \end{align}
> hold. These inequalities indicate that the global depolarizing channel is much weaker than local noise, given the same noise strength. Consequently, for the global depolarizing channel, our algorithm is capable of handling scenarios where $p_g\leq\mathcal{O}(1)$, while maintaining comparable prediction accuracy. This suggests that the algorithm can effectively handle both types of noise, with only a minor difference in performance under different noise strengths. **We have included this detailed discussion in the revised manuscript on Pages 16-17.**

---

> ### Author Response · Authors · 2024-11-20
> **Addressing "Questions/Weaknesses in Official Review of 3076 by Reviewer wYLf" (part 2)**
>
> **Comment 3**: *The paper should include a more detailed discussion of the computational and practical limitations of the algorithm (for example, stress-testing the algorithm under increased complexities and reporting on its degradation or stability performance.*
>
> **Response**: Thank you for this insightful comment. We will include following discussions to the revised manuscript.
>
> **Theoretical Guarantee**:
> The time complexity and sample complexity of our algorithm are rigorously derived in Theorems 2 and 3, both of which depend polynomially on the number of qubits of the target quantum state. These results imply that our learning algorithm remains efficient even as the complexity of the input states increases.
>
> **Numerical Simulation**:
> we supplemented the stress testing of the proposed quantum learning algorithm in this study. Specifically, we applied the algorithm to estimate the complexity lower bound of quantum states under noisy environments for one-dimensional dynamical evolution. We tested the transverse-field Ising model with system sizes of $1\times 6$, $1\times 8$, and $1\times 10$, and the numerical results are presented in Figure 3. Our findings indicate that the complexity lower bound of noisy quantum circuits does not grow linearly with circuit depth, which is consistent to the results given by Figure 1 in the main file.
>
> Furthermore, we observed that as the system size and complexity increase, the curves corresponding to two different quantum states, $\tilde{R}=2$ and $\tilde{R}=10$, exhibit similar overall trends for the system size $n\in\{6, 8, 10\}$. Moreover, the variance of the algorithm does not increase with the growing system size but remains consistently stable. At the same time, the gap between the curves of quantum states with different complexities does not show a closing trend as the system size grows. These characteristics demonstrate the robustness of our learning algorithm against increases in system size and complexity. Therefore, our algorithm remains reliable for predicting the complexity of large-scale weakly noisy quantum states. **Please refer to Appendix L on Pages 28-19.**
>
> **Comment 4**: *Incorporating key proofs and insights from the appendices into the main body would enhance the paper's value, as some essential concepts, like the subroutine BMaxS, are only briefly mentioned in the main text.
> It would also be very useful if the manuscript provided a framework for empirical testing and validation of the algorithm, including benchmark selection and metrics of validation. This would help bridge the missing gap between theoretical research and practical applications and give a much stronger justification for the effectiveness of the algorithm in noisy quantum environments.*
>
> **Response:** We thank the reviewer for their valuable comment. In response, we have clarified several related concepts in the revised manuscript. Due to the 10-page limit for the main text, most of the details about the BMaxS algorithm are provided in the supplementary material, but we have made every effort to explain the key aspects of the algorithm within the main text. Additionally, we have included a numerical simulation in the revised manuscript to analyze the Hamiltonian dynamics from Ref.[Youngseok Kim et al., 2023] and benchmark the complexity of noisy Hamiltonian dynamics. Our simulations are capable of testing the algorithm on up to a 12-qubit density matrix, which corresponds to the complexity of a 24-qubit state vector.
> **Please refer to Page 7 for explaining BMaxS, and Pages 8-9 and Pages 28-29 for supplemented numerical simulation results.**
>
> Youngseok Kim, Andrew Eddins, Sajant Anand, KenXuan Wei, Ewout van den Berg, Sami Rosenblatt, Hasan Nayfeh, Yantao Wu, Michael Zaletel, Kristan Temme, and Abhinav Kandala. Evidence for the utility of quantum computing before fault tolerance. Nature, 618:500–505, 2023.

---

> ### Author Response · Authors · 2024-11-20
> **Concluding Remarks**
>
> We thank the reviewer for their insightful comments and suggestions. In response, we have clarified the quantum hardware requirements of our learning algorithm and highlighted its robustness under both local and global noise channels. To validate the practical applicability of our algorithm, we conducted numerical simulations and stress tests. The results successfully demonstrate that the complexity of weakly noisy quantum states exhibits a fundamentally different trend compared to that of pure random quantum states, thereby broadening the understanding of quantum state complexity in the regime of weak noise. We hope these revisions and clarifications address your concerns and improve the clarity of our arguments. We trust that these changes will help in reassessing the evaluation of our work.

---

> ### Author Response · Authors · 2024-11-22
>
> Dear Reviewer wYLf,
>
> Thanks agian for your constructive suggestions. We have made our best effort to address your comments and revised the manuscript accordingly. As the public discussion period is nearing its end, we would be delighted to hear your feedback and suggestions on our latest responses. We look forward to your reply.
>
> Best regards!

---

> > ### Comment · Reviewer_wYLf · 2024-11-22
> >
> > Thank you for your comprehensive clarifications and the additional experiments. You have addressed nearly all of my concerns, and I will update my rating accordingly.

---

> > > ### Author Response · Authors · 2024-11-22
> > >
> > > We greatly appreciate your kind response and are delighted that our revisions have met your satisfaction！

---

### Official Review · Reviewer_eyDN · 2024-10-31

**Soundness:** 3
**Presentation:** 2
**Contribution:** 2
**Rating:** 6
**Confidence:** 3

**Summary:**

The paper proposes a quantum learning algorithm to predict the complexity of weakly noisy quantum states. The authors introduce the concepts of Weakly Noisy Quantum State and Limited-Structured Complexity and prove that their algorithm achieves optimal sample complexity with respect to their definition of noisy circuit depth. They also provide a lower bound for sample complexity. This work also shows meaningful connections between learning algorithms and quantum state complexity.

**Strengths:**

The paper studies a fundamental problem of understanding the learning complexity of noisy quantum states, which has potential applications in various fields claimed for black-hole theory and condensed-matter physics. The authors propose a novel quantum learning algorithm that leverages the intrinsic structure of quantum circuit architectures to predict the complexity of weakly noisy quantum states. The theoretical analysis provides provable guarantees on the sample complexity and efficiency of the proposed algorithm.

**Weaknesses:**

- The setting of weakly noisy quantum states is not well-motivated, and it is unclear how the proposed learning approach can be feasible in practice. The accumulation of quantum noise on each quantum system can be significant, and the considered noise model may not be sufficiently practical or meaningful. Not sure about the the noise channel is assumed to be gate-independent.
- The paper is difficult to follow, with various parameters in the theorems that are not clearly explained (e.g., Theorem 4). The use of the same letter R to define both the noise numbers in the definition of Weakly Noisy Quantum States and the regrets in later sections adds to the confusion.
- The paper does not provide detailed examples or numerical experiments to support the proposed algorithm's effectiveness and practicality. This lack of empirical evidence makes the results less convincing.
- It is not very clear to me whether the main algorithms consider quantum noises.
- It is not convincing that the results are of a broad interest in the ICLR community.
- The results presented in the paper do not appear to be particularly surprising or groundbreaking in the field of quantum learning theory.

**Questions:**

- Could you provide a more compelling motivation for studying weakly noisy quantum states? How do they relate to real-world quantum systems, and what are the potential applications of your findings?
- Given that quantum noise can accumulate rapidly in depth significantly in each quantum system, how does your proposed learning approach remain feasible in practice? Why gate-independent noise channel? Please clarify the practicality of your noise model and its implications for the scalability of your method. Better to compare with an explicit circuit model example.
- Several parameters in the theorems, such as Theorem 4, are not clearly explained. Could you provide more context and explanations for these parameters to improve the readability of the paper?
- The use of the same letter R to define both the noise numbers in the definition of Weakly Noisy Quantum States and the regrets in later sections is confusing. Please consider using different notations for clarity and consistency throughout the paper.
- To support the effectiveness and practicality of your proposed algorithm, could you provide detailed examples that illustrate its workings and benefits?
- Do you assume quantum noises in the main algorithm 1?
- Numerical experiments on benchmark datasets or simulated quantum systems would strengthen the empirical evidence for your method. Could you include such experiments to demonstrate the performance of your algorithm in various settings?
- While your paper introduces a new quantum learning algorithm, it would be helpful to provide a more comprehensive comparison with existing methods in the field of quantum learning theory. Could you discuss how your approach differs from and improves upon prior work?
- Could you elaborate on how your work on learning the complexity of weakly noisy quantum states is relevant to the broader ICLR community?

---

> ### Author Response · Authors · 2024-11-20
> **Addressing "Questions/Weaknesses in Official Review of 3076 by Reviewer eyDN" (part 1)**
>
> **Comment 1**: *Could you provide a more compelling motivation for studying weakly noisy quantum states? How do they relate to real-world quantum systems, and what are the potential applications of your findings?*
>
> **Response**: Thanks for this valuable suggestion. In our paper, weakly noisy quantum states refer to quantum states with $\mathcal{O}(\log n)$ depth, where each quantum gate is affected by a $p=\mathcal{O}(n^{-1})$-strength noise channel. We note that weakly noisy quantum states are ubiquitous in the real world. For example, in the current Noisy Intermediate-Scale Quantum (NISQ) era, quantum computers generally have $\leq 10^{3}$ qubits, and the noise strength of single- and double-qubit gates is generally $10^{-3}$. Due to the limited coherent time of quantum devices, NISQ quantum computers can only process shallow (constant or logarithmic) depth quantum circuits. As a result, all quantum states prepared by near-term quantum computers can be recognized as weakly noisy quantum states.
>
>
> Understanding the complexity of weakly noisy states have several significant applications. Firstly, the predicted complexity enables the classification of the target unknown state $\rho_{\rm un}$ into one of the following scenarios: (i) $\rho_{\rm un}$ can be approximated by an estimator generated by constant-depth noiseless circuits, which is amenable to classical simulation as shown in [J. C. Napp et al., 2022, S. Bravyi et al., 2021], (ii)  $\rho_{\rm un}$ can be approximated using noiseless circuits with sub-logarithmic depth, a task that may present classical computational challenges, as discussed in [A. Deshpande et al., 2022], and (iii) $\rho_{\rm un}$ cannot be approximated by any linear combination of circuits constrained to a specific architecture with a depth of at most $\log n$. Besides benchmarking the NISQ computational power (whether its output can be classically simulated), the proposed quantum algorithm and predicted weakly noisy state complexity may have practical applications in various fields, including understanding the black hole theory and the quantum phase transition. For example, in the context of the Anti-de-Sitter-space/Conformal Field Theory (AdS/CFT) correspondence, the ``complexity equals volume'' conjecture [D. Stanford et al., 2014] suggests that the boundary state of the correspondence has a complexity proportional to the volume behind the event horizon of a black hole in the bulk geometry. However, when measuring the boundary state, the interaction with the surrounding environment inevitably introduces a noise signal, making the pure state noisy. Furthermore, in quantum many-body systems, quantum phase transitions occur when the external parameters varies [S. Sachdev 1999], and the ability to correctly predict the quantum phase transition boundary can help us understand many strong-correlated systems [B. Zheng et al., 2017]. It is known that quantum topological phases can be distinguished by their ground state complexity [Y. Huang et al., 2015], and the shadow tomography [H. Huang et al., 2022] method utilized quantum channels to provide a noisy state approximation to the ground state. Predicting complexity of such noisy ground state approximations may recognize the topological phases of matter. **We have supplemented above discussions in the revised manuscript on Page~2.**
>
> References:
>
> John C Napp, Rolando L La Placa, Alexander M Dalzell, Fernando GSL Brandao, and Aram W Harrow.
> Efficient classical simulation of random shallow 2d quantum circuits. Physical Review X, 12(2):021021, 2022.
>
> Sergey Bravyi, David Gosset, and Ramis Movassagh. Classical algorithms for quantum mean values. Nature
> Physics, 17(3):337–341, 2021.
>
> Abhinav Deshpande, Pradeep Niroula, Oles Shtanko, Alexey V Gorshkov, Bill Fefferman, and Michael J
> Gullans. Tight bounds on the convergence of noisy random circuits to the uniform distribution. PRX
> Quantum, 3(4):040329, 2022.
>
> Douglas Stanford and Leonard Susskind. Complexity and shock wave geometries. Physical Review D,
> 90(12):126007, 2014.
>
> Subir Sachdev. Quantum phase transitions. Physics World, 12(4):33, 1999.
>
> Bo-Xiao Zheng, Chia-Min Chung, Philippe Corboz, Georg Ehlers, Ming-Pu Qin, Reinhard M Noack, Hao
> Shi, Steven R White, Shiwei Zhang, and Garnet Kin-Lic Chan. Stripe order in the underdoped region of
> the two-dimensional hubbard model. Science, 358(6367):1155–1160, 2017.
>
> Yichen Huang, Xie Chen, et al. Quantum circuit complexity of one-dimensional topological phases. Physical
> Review B, 91(19):195143, 2015.
>
> Hsin-Yuan Huang, Richard Kueng, Giacomo Torlai, Victor V Albert, and John Preskill. Provably efficient
> machine learning for quantum many-body problems. Science, 377(6613):eabk3333, 2022.

---

> ### Author Response · Authors · 2024-11-20
> **Addressing "Questions/Weaknesses in Official Review of 3076 by Reviewer eyDN" (part 2)**
>
> **Comment 2**: *Given that quantum noise can accumulate rapidly in depth significantly in each quantum system, how does your proposed learning approach remain feasible in practice? Why gate-independent noise channel? Please clarify the practicality of your noise model and its implications for the scalability of your method. Better to compare with an explicit circuit model example.*
>
> **Response**: We thank the reviewer for these comments. In the proposed learning approach (Alg~1), the algorithm requires (i) the target weakly noisy quantum state $\rho_{\rm un}$, and (ii) the QCA state set $\hat{\rho}_{QCA}$.
>
> The target weakly noisy state naturally contains noise signals, while the QCA state set $\hat{\rho}_{\rm QCA}$ is provided by the classical shadow representation [H. Huang et al., 2020]. When preparing the classical shadow, recent result successfully embedded the quantum error mitigation method into the classical shadow protocol, which provides rigorous theoretical guarantees even in the noisy environment [H. Jnane et al., 2024].
>
>
> In quantum computing research, the gate-independent noise model assumes that the noise affecting quantum operations is uniform across all gates, regardless of their type or implementation. This simplification is widely adopted for several reasons:
>
> **Theoretical Simplification:** Assuming gate-independent noise allows researchers to develop and analyze error correction protocols and fault-tolerant methods without delving into the complexities introduced by gate-specific noise characteristics [E. Knill et al., 2008, J. Helsen et al., 2019, J. Claes et al., 2021, S. Chen et al., 2021]. This uniformity facilitates the derivation of general results and theoretical bounds [M. A Nielsen et al., 2001].
>
> **Practical Approximations:** In certain quantum systems, particularly those with well-calibrated gates operating on the same number of qubits and utilizing uniform control mechanisms, noise variations across different gates can be negligible [P. W. Shor, 1996, F. Arute et al., 2019]. In such cases, the gate-independent noise model serves as a reasonable approximation, streamlining analysis without significantly compromising accuracy.
>
> **Alignment with Twirled Noise Models:**
> Techniques like Pauli twirling are employed to transform complex noise channels into diagonal forms on the Pauli basis [J. Wallman et al., 2016]. While twirling simplifies the noise structure, it does not inherently eliminate gate dependence. However, in many scenarios, the resulting noise can be approximated as gate-independent, aligning with the assumptions of the model.
>
> The gate-independent noise model provides a foundational framework for understanding error propagation and developing correction strategies, which is a useful abstraction for theoretical exploration and the initial development of error correction methods. We leave an open question of how to depict the gate-dependent noise, which usually happens in larger, more complex quantum architectures. **We have supplemented above discussions to Page 16. For explicit circuit model example, please refer to Sec. 6 on Page 9.**
>
> References:
>
> Hsin-Yuan Huang, Richard Kueng, and John Preskill. Predicting many properties of a quantum system
> from very few measurements. Nature Physics, 16(10):1050–1057, 2020.
>
> Hamza Jnane, Jonathan Steinberg, Zhenyu Cai, H Chau Nguyen, and B´alint Koczor. Quantum error
> mitigated classical shadows. PRX Quantum, 5(1):010324, 2024.
>
> Emanuel Knill, Dietrich Leibfried, Rolf Reichle, Joe Britton, R Brad Blakestad, John D Jost, Chris Langer,
> Roee Ozeri, Signe Seidelin, and David J Wineland. Randomized benchmarking of quantum gates. Physical
> Review A, 77(1):012307, 2008.
>
> Jonas Helsen, Xiao Xue, Lieven MK Vandersypen, and Stephanie Wehner. A new class of efficient random-
> ized benchmarking protocols. npj Quantum Information, 5(1):1–9, 2019.
>
> Jahan Claes, Eleanor Rieffel, and Zhihui Wang. Character randomized benchmarking for non-multiplicity-
> free groups with applications to subspace, leakage, and matchgate randomized benchmarking. PRX Quan-
> tum, 2(1):010351, 2021.
>
> Senrui Chen, Wenjun Yu, Pei Zeng, and Steven T. Flammia. Robust shadow estimation. PRX Quantum,
> 2:030348, Sep 2021.
>
> Michael A Nielsen and Isaac L Chuang. Quantum computation and quantum information, volume 2. Cam-
> bridge university press Cambridge, 2001.
>
> Peter W Shor. Fault-tolerant quantum computation. In Proceedings of 37th conference on foundations of
> computer science, pages 56–65. IEEE, 1996.
>
> Frank Arute, Kunal Arya, Ryan Babbush, Dave Bacon, Joseph C Bardin, Rami Barends, Rupak Biswas,
> Sergio Boixo, Fernando GSL Brandao, David A Buell, et al. Quantum supremacy using a programmable
> superconducting processor. Nature, 574(7779):505–510, 2019.
>
> Joel J Wallman and Joseph Emerson. Noise tailoring for scalable quantum computation via randomized
> compiling. Physical Review A, 94(5):052325, 2016.

---

> > ### Author Response · Authors · 2024-11-20
> > **Addressing "Questions/Weaknesses in Official Review of 3076 by Reviewer eyDN" (part 3)**
> >
> > **Comment 3**: *Several parameters in the theorems, such as Theorem 4, are not clearly explained. Could you provide more context and explanations for these parameters to improve the readability of the paper?*
> >
> > **Response**: We appreciate the reviewer’s feedback and have ensured that all relevant parameters are now clearly defined and explained. **Please refer to the revised Theorem 4 on Page 8.**
> >
> > Specifically, Theorem 4 specifies the following:
> >
> > **1** $\rho_{\rm un}$ denotes the target weakly noisy state;
> >
> > **2** $\hat{\rho}_{\rm QCA}(R,\mathcal{A},N)$ represents a set of random quantum states generated by an $R$-depth quantum circuit constrained by the architecture $\mathcal{A}$;
> >
> > **3** The Boolean function $\mathcal{P}(R)$, formally defined in Eq. 14 of the main text, characterizes whether the complexity of the target quantum state satisfies $C(\rho_{\rm un}) \leq LR$ or $C(\rho_{\rm un})>LR$.
> >
> > These clarifications have been incorporated into the revised manuscript.
> >
> >
> > **Comment 4**: *The use of the same letter R to define both the noise numbers in the definition of Weakly Noisy Quantum States and the regrets in later sections is confusing. Please consider using different notations for clarity and consistency throughout the paper.*
> >
> > **Response**: Thanks for your comment, we have changed a different notation to represent the average regret. **Please also refer to the revised Theorem 4 on Page 8.**
> >
> > **Comment 5**: *To support the effectiveness and practicality of your proposed algorithm, could you provide detailed examples that illustrate its workings and benefits? Numerical experiments on benchmark datasets or simulated quantum systems would strengthen the empirical evidence for your method. Could you include such experiments to demonstrate the performance of your algorithm in various settings?*
> >
> > **Response**: We thank the reviewer for these comments. We have supplemented a numerical simulation to learn from the Hamiltonian dynamics studied in [Y. Kim et al., 2023] and benchmark the complexity of noisy Hamiltonian dynamics. We conduct a numerical evaluation of our algorithm across various configurations of the transverse-field Ising model, encompassing both one-dimensional and two-dimensional scenarios. Our simulations incorporate diverse noisy circuit depths, while scaling to accommodate a 12-qubit density matrix—an equivalent complexity to that of a 24-qubit state vector. **Please refer to the Sec 6 and Appendix L of the revised manuscript for numerical simulation results.**Numerical simulation highlights *weakly noisy quantum state complexity lower bound may not grow with the circuit depth, that is dramatically different to that of pure states.*
> >
> > Youngseok Kim, Andrew Eddins, Sajant Anand, KenXuan Wei, Ewout van den Berg, Sami Rosenblatt, Hasan Nayfeh, Yantao Wu, Michael Zaletel, Kristan Temme, and Abhinav Kandala. Evidence for the utility of quantum computing before fault tolerance. Nature, 618:500–505, 2023.
> >
> > **Comment 6**: *Do you assume quantum noises in the main algorithm 1?*
> >
> > **Response**: We appreciate the reviewer’s comment. As explained in response to Comment 2, we assume that the quantum state under investigation is weakly noisy, characterized by gate-independent noise that can be described using a set of Kraus operators. In the proposed learning approach (Algorithm 1), the method requires: (i) the target weakly noisy quantum state $\rho_{\rm un}$,
> > and (ii) the QCA state set $\hat{\rho}_{QCA}$.
> >
> > The target weakly noisy state naturally contains noise signals, while the QCA state set $\hat{\rho}_{QCA}$
> >
> > is provided by the classical shadow representation [H. Huang et al., 2020]. For the preparation of the classical shadow, an ideal scenario would involve access to a fault-tolerant quantum computer capable of error correction. As noted in response to Comment 2, recent work has successfully integrated quantum error mitigation techniques into the classical shadow protocol, offering rigorous theoretical guarantees even in noisy environments [S. Chen et al., 2021, H. Jnane et al., 2024]. Consequently, in Algorithm 1, the studied weakly noisy state is assumed to be noisy, while the classical shadow set $\hat{\rho}_{QCA}$ is treated as noise-free.
> >
> > References:
> >
> > Hsin-Yuan Huang, Richard Kueng, and John Preskill. Predicting many properties of a quantum system from very few measurements. Nature Physics, 16(10):1050–1057, 2020.
> >
> > Hamza Jnane, Jonathan Steinberg, Zhenyu Cai, H Chau Nguyen, and B´alint Koczor. Quantum error
> > mitigated classical shadows. PRX Quantum, 5(1):010324, 2024.
> >
> > Senrui Chen, Wenjun Yu, Pei Zeng, and Steven T. Flammia. Robust shadow estimation. PRX Quantum,
> > 2:030348, Sep 2021.

---

> > > ### Author Response · Authors · 2024-11-20
> > > **Addressing "Questions/Weaknesses in Official Review of 3076 by Reviewer eyDN" (part 4)**
> > >
> > > **Comment 8**: *While your paper introduces a new quantum learning algorithm, it would be helpful to provide a more comprehensive comparison with existing methods in the field of quantum learning theory. Could you discuss how your approach differs from and improves upon prior work?*
> > >
> > > **Response:**
> > > We thank the reviewer for the valuable suggestion. We note that the task of learning the complexity of quantum weakly noisy states is fundamentally a novel challenge that bridges learning theory and computational complexity. To the best of our knowledge, this topic has received limited attention in existing literature. Nonetheless, we aim to provide a review of related works on quantum state and circuit learning, highlighting their connections and distinctions with our approach.
> > >
> > >
> > > In the context of condensed matter physics, a topological phase transition occurs when the ground states of a family of Hamiltonians exhibit different circuit complexities. This suggests that topological phase classification may help distinguish between low-complexity and high-complexity ground states. The use of learning algorithms to classify quantum phases of matter has been widely studied. Proposals include quantum neural networks [I. Cong et al., 2019], classical neural networks [M. Beach et al., 2018, J. Carrasquilla et al., 2017, E. Greplova et al., 2020, F. Schindler et al., 2017, E. V. Nieuwenburg et al., 2017], and other classical machine learning models [H. Huang et al., 2022, J. F. Rodriguez-Nieva et al., 2019]. However, most previous works lack rigorous theoretical guarantees. Among these studies, H. Huang et al., 2022 utilized shadow tomography of the concerned ground states to design an unsupervised machine learning approach guaranteed to classify accurately under certain conditions. The proposed ``shadow kernel" can generate quantum entropies by tuning hyperparameters, enabling the approximation of various topological phase order parameters. Compared with[H. Huang et al., 2022], our learning algorithm can be applied to noisy states and predict complexity, whereas [H. Huang et al., 2022] mainly focuses on pure ground state classification without providing an explicit metric for characterizing complexity.
> > >
> > > On the other hand, learning quantum states and circuits is a long-standing task in the field of quantum machine learning. Previous works generally utilized parameterized circuits to learn approximations of target states and circuits [Kosuke Mitarai et al., 2018, Feidiao Yang et al., 2020, Hsin-Yuan Huang et al., 2021, Sofiene Jerbi et al. 2021, M Cerezo et al., 2022]. However, variational-based methods generally lack theoretical guarantees and may suffer from the barren plateau phenomenon even in low-depth parameterized circuits [Jarrod R McClean et al., 2018, Marco Cerezo, et al. 2021, Eric R Anschuetz et al. 2022]. Very recently, classical shadow-based learning algorithms have been proposed for reconstructing shallow quantum circuits  $U$ [H. Huang et al., 2024, Z. Landau et al., 2024]. Nevertheless, these methods require querying $U^{\dagger}$, which introduces intrinsic limitations in noisy environments since most noisy channels (CPTP maps) may not be invertible. Compared with related works, our method bypasses these limitations by introducing the "Intrinsic Structure property" (as given by Theorem 1), which enables an efficient quantum learning algorithm to predict state complexity in noisy environments.
> > >
> > >
> > > **We have supplemented above comparisons into the revised manuscript. Please refer to Appendix. A on Page 15.**
> > >
> > > References:
> > > Iris Cong, et al. Quantum convolutional neural networks. Nature Physics,
> > > 15(12):1273–1278, 2019.
> > >
> > > Matthew JS Beach, et al. Physical Review B, 97(4):045207, 2018.
> > >
> > > Juan Carrasquilla et al.  Nature Physics, 13(5):431–434,
> > > 2017.
> > >
> > > Eliska Greplova et al., New Journal of Physics, 22(4):045003, 2020.
> > >
> > > Frank Schindler, et al., Physical Review B, 95(24):245134, 2017.
> > >
> > > Evert PL Van Nieuwenburg, et al., Nature Physics, 13(5):435–439, 2017.
> > >
> > > Hsin-Yuan Huang et al., Science, 377(6613):eabk3333, 2022.
> > >
> > > Joaquin F Rodriguez-Nieva et al., Nature Physics, 15(8):790–795, 2019.
> > >
> > > Kosuke Mitarai et al., Physical Review A, 98(3):032309, 2018.
> > >
> > > Feidiao Yang et al., In Proceedings of the AAAI Conference on Artificial Intelligence, volume 34, pages 6607–6614, 2020.
> > >
> > > Hsin-Yuan Huang et al.,Nature Communications, 12(1):1–9,
> > > 2021.
> > >
> > > Sofiene Jerbi et al., arXiv preprint arXiv:2110.13162, 2021.
> > >
> > > M Cerezo et al., Nature Computational Science, 2(9):567–576, 2022.
> > >
> > > Jarrod R McClean et al.,  Nature Communications, 9(1):1–6, 2018.
> > >
> > > Marco Cerezo, et al., Nature Communications, 12(1):1–12, 2021.
> > >
> > > Eric R Anschuetz et al., Nature Communications, 13(1):7760, 2022.
> > >
> > > Hsin-Yuan Huang, et al. arXiv preprint arXiv:2401.10095, 2024.
> > >
> > > Zeph Landau et al. arXiv preprint arXiv:2410.23618, 2024.

---

> > > > ### Author Response · Authors · 2024-11-20
> > > > **Addressing "Questions/Weaknesses in Official Review of 3076 by Reviewer eyDN" (part 5)**
> > > >
> > > > **Comment 9**: *Could you elaborate on how your work on learning the complexity of weakly noisy quantum states is relevant to the broader ICLR community?*
> > > >
> > > > **Response**: The core focus of the ICLR conference lies in advancing learning and representation, with two pivotal themes: the representation of outputs or states and applications within physics. This is further underscored by the 2024 Nobel Prize in Physics, which was awarded for contributions to machine learning, highlighting the profound impact of learning algorithms in uncovering the fundamental nature of the physical world.
> > > >
> > > > In the realm of quantum science, the complexity of quantum states and circuits plays a central role in quantum information, quantum computing, and quantum many-body systems, representing intrinsic properties of the quantum domain. However, rigorously characterizing this complexity has remained a significant challenge [J. Haferkamp et al., 2022]. For noiseless systems, Ref. [T. Schuster et al., 2024] established that predicting the complexity ${\rm poly}\log n$-depth random quantum circuits is a difficult task, even for quantum learning algorithms. Building on this, our work successfully extends these insights to noisy environments. Specifically, we demonstrate that ${\rm poly}\log n$-depth weakly noisy quantum states have fundamentally different statistical properties compared to noiseless quantum states, and we have proposed an algorithm to effectively learn the complexity of weakly noisy quantum states.
> > > > Moreover, we prove that our algorithm achieves `optimal' sample complexity concerning circuit depth, overcoming the training limitations of previous works  [Kosuke Mitarai et al., 2018, Feidiao Yang et al., 2020, Hsin-Yuan Huang et al., 2021, Sofiene Jerbi et al. 2021, M Cerezo et al., 2022] and extends previous arts into the noisy environment [H. Huang et al., 2024, Z. Landau et al., 2024].
> > > >
> > > > We believe that exploring the properties of weakly noisy quantum circuits is a significant and valuable area of research for the ICLR conference. **Below, we highlight several existing ICLR papers that leverage the characteristics of shallow noisy quantum circuits:**
> > > >
> > > > Ref. [Ismail Yunus Akhalwaya et al., 2024] presents NISQ-TDA, an innovative end-to-end quantum machine learning algorithm tailored for high-dimensional classical data. Its design emphasizes short circuit depths, making it well-suited for noisy intermediate-scale quantum (NISQ) devices.
> > > >
> > > > Ref. [Yash J. Patel et al., 2024] proposes a curriculum-based reinforcement learning approach to optimize quantum circuit architectures in the presence of noise, addressing practical challenges in noisy environments.
> > > >
> > > > Ref. [Jonas Landman et al., 2023] explores the capability of classical methods to approximate the performance of variational quantum circuits (VQCs) in machine learning tasks, shedding light on the interplay between classical and quantum approaches.
> > > >
> > > > **These studies highlight the growing interest in understanding and leveraging noisy quantum systems within the machine learning community.**
> > > >
> > > > In summary, this breakthrough connects the important topics of machine learning and noisy quantum state complexity. We believe our results align well with the themes of ICLR and will be widely appealing to its readers. We hope these revisions and clarifications address your concerns and improve the clarity of our arguments. We trust that these changes will help in reconsidering the evaluation of our work. Thank you again for your thoughtful feedback.
> > > >
> > > > References:
> > > >
> > > > Jonas Haferkamp et al., Nature Physics, pages 1–5, 2022.
> > > >
> > > > Thomas Schuster et al., arXiv preprint arXiv:2407.07754, 2024.
> > > >
> > > > Kosuke Mitarai et al., Physical Review A, 98(3):032309, 2018.
> > > >
> > > > Feidiao Yang et al., In Proceedings of the AAAI Conference on Artificial Intelligence, volume 34, pages 6607–6614, 2020.
> > > >
> > > > Hsin-Yuan Huang et al.,Nature Communications, 12(1):1–9, 2021.
> > > >
> > > > Sofiene Jerbi et al., arXiv preprint arXiv:2110.13162, 2021.
> > > >
> > > > M Cerezo et al., Nature Computational Science, 2(9):567–576, 2022.
> > > >
> > > > Jarrod R McClean et al., Nature Communications, 9(1):1–6, 2018.
> > > >
> > > > Marco Cerezo, et al., Nature Communications, 12(1):1–12, 2021.
> > > >
> > > > Eric R Anschuetz et al., Nature Communications, 13(1):7760, 2022.
> > > >
> > > > Hsin-Yuan Huang, et al. arXiv preprint arXiv:2401.10095, 2024.
> > > >
> > > > Zeph Landau et al. arXiv preprint arXiv:2410.23618, 2024.
> > > >
> > > > Ismail Yunus Akhalwaya et al., In The Twelfth International Conference on Learning Representations, ICLR 2024,
> > > > Vienna, Austria, May 7-11, 2024. OpenReview.net, 2024.
> > > >
> > > > Yash J. Patel et al., In TheTwelfth International Conference on Learning Representations, ICLR 2024, Vienna, Austria, May 7-11,
> > > > 2024. OpenReview.net, 2024.
> > > >
> > > > Jonas Landman et al., In The Eleventh International Conference on Learning Representations, ICLR 2023, Kigali, Rwanda, May 1-5, 2023. OpenReview.net,
> > > > 2023.

---

> ### Author Response · Authors · 2024-11-22
>
> Dear Reviewer eyDN,
>
> Thanks agian for your constructive suggestions. We have made our best effort to address your comments and revised the manuscript accordingly. As the public discussion period is nearing its end, we would be delighted to hear your feedback and suggestions on our latest responses. We look forward to your reply.
>
> Best regards!

---

> > ### Comment · Reviewer_eyDN · 2024-11-25
> >
> > Thank you for your clarifications. Most of my concerns have been addressed, and I will update my rating.

---

> > > ### Author Response · Authors · 2024-11-25
> > >
> > > We would like to express our gratitude again for your valuable suggestions, which have been very important in improving our paper. Finally, thank you for your satisfaction with our revisions and responses！

---

### Official Review · Reviewer_m7qv · 2024-11-03

**Soundness:** 2
**Presentation:** 3
**Contribution:** 3
**Rating:** 6
**Confidence:** 4

**Summary:**

This paper investigates a fundamental question in quantum state complexity: how to predict/learn the complexity of weakly noisy quantum states. The authors develop an efficient learning algorithm that:

1. Uses classical shadow representation of quantum states
2. Provides optimal sample complexity with polynomial classical processing time
3. Works specifically for weakly noisy states (noise strength O(1/n) and depth O(poly log n))

The main contribution is establishing an efficient algorithm for learning quantum state complexity in noisy environments.

**Strengths:**

- The algorithm is rigorously analyzed with provable guarantees on both sample and computational complexity.

- The work studies quantum state complexity in noisy environments which better reflect real quantum devices.

- The approach cleverly leverages classical shadows and intrinsic properties of quantum circuit architectures to make the learning problem tractable.

- The sample complexity is shown to be optimal with respect to circuit depth.

**Weaknesses:**

- It is well known in quantum complexity theory that the circuit complexity of a quantum state cannot be efficiently learned [1, 2, 3]. For example, the existence of pseudorandom states that can be generated in polylog depth on 1D circuits [3] immediately implies that no polynomial-time quantum algorithm can distinguish between polylog circuit complexity (complexity defined in terms minimum circuit depth) and exponential circuit complexity. Hence, it is not possible to predict complexity of a state in polynomial processing time without first solving quantumly hard cryptographic problems. I find it very confusing that the authors claim to predict circuit complexity in polynomial time.

- Despite the focus of practical importance (the presence of noise), the paper did not provide any numerical experiments to validate the theoretical claims (including both efficient sample and computational complexity). I think it is important for this work to have supporting numerical experiments for system size that scales to 50-100 qubits.

- While the paper focuses on weakly noisy states, it's unclear how the approach would scale to more general noise models or stronger noise regimes. I think the work did not provide enough justification for the focus on weakly noisy states.

[1] "Pseudorandom quantum states." Advances in Cryptology–CRYPTO 2018.
[2] "Quantum pseudoentanglement." arXiv preprint arXiv:2211.00747 (2022).
[3] "Random unitaries in extremely low depth." arXiv preprint arXiv:2407.07754 (2024).

**Questions:**

Could the authors provide further clarification on how it is possible to predict the circuit complexity of a state in polynomial time? There are a plethora of existing results proving that circuit complexity is not efficiently learnable. Assuming the claims given in this work are correct, could the authors provide a detailed exposition for how this seemingly contradictory statements can be resolved?

Could the authors provide numerical simulations demonstrating their learning algorithm's performance on concrete examples of noisy quantum states? This would help validate the theoretical guarantees and provide intuition about the practical performance.

The paper focuses on weakly noisy states with specific noise parameters (O(1/n) strength and O(poly log n) depth). Could the authors comment on whether similar techniques could be extended to more general noise models or stronger noise regimes? If not, could the authors describe if stronger results are simply impossible for any learning algorithm?

---

> ### Author Response · Authors · 2024-11-20
> **Addressing "Questions/Weaknesses in Official Review of 3076 by Reviewer m7qv" (part 1)**
>
> **Comment 1**: *It is well known in quantum complexity theory that the circuit complexity of a quantum state cannot be efficiently learned [1, 2, 3]. For example, the existence of pseudorandom states that can be generated in polylog depth on 1D circuits [3] immediately implies that no polynomial-time quantum algorithm can distinguish between polylog circuit complexity (complexity defined in terms minimum circuit depth) and exponential circuit complexity. Hence, it is not possible to predict complexity of a state in polynomial processing time without first solving quantumly hard cryptographic problems. I find it very confusing that the authors claim to predict circuit complexity in polynomial time. Could the authors provide further clarification on how it is possible to predict the circuit complexity of a state in polynomial time? There are a plethora of existing results proving that circuit complexity is not efficiently learnable. Assuming the claims given in this work are correct, could the authors provide a detailed exposition for how this seemingly contradictory statements can be resolved?*
> [1] "Pseudorandom quantum states." Advances in Cryptology–CRYPTO 2018. [2] "Quantum pseudoentanglement." arXiv preprint arXiv:2211.00747 (2022). [3] "Random unitaries in extremely low depth." arXiv preprint arXiv:2407.07754 (2024).
>
> **Response**:
> We truly appreciate the reviewer for mentioning very recent work [T. Schuster et al., 2024]. In our paper, we primarily focus on the task of learning specific quantum states using shallow-depth circuits. This approach does not contradict the concept of *computational indistinguishability* between two quantum random ensembles. To elaborate, a pseudorandom unitary (PRU) refers to a family of efficiently computable quantum circuits that ensure no polynomial-time quantum algorithm can distinguish between a unitary drawn from the PRU family and one sampled from the Haar measure, in other words, learning the difference between two ensembles is quantum hard. In contrast, our learning problem involves reconstructing a given shallow-depth, weakly noisy quantum state by optimizing the a loss function related to the original state and the reconstructed state. Furthermore, recent works [H. Huang et al., 2024, Z. Landau et al., 2024] have also demonstrated that fixed shallow quantum circuits and quantum pure states (including $1$D-$\mathcal{O}(\log n)$-depth and all-to-all $\mathcal{O}(\log\log n)$-depth) can be efficiently learned. These results demonstrate significant differences between scenarios with and without 'randomness'. Compared with [T. Schuster et al., 2024], our work is more similar to Refs.[H. Huang et al., 2024, Z. Landau et al. 2024], which focus on a specific quantum state rather than an ensemble. We also note that the pure state can be classified according to their complexity when the target states exhibit certain physical structures, such as the ground state of XXZ models and the toric code, as studied in Ref.~[H. Huang et al., 2022]
>
> References:
>
> Thomas Schuster, Jonas Haferkamp, and Hsin-Yuan Huang. Random unitaries in extremely low depth. arXiv preprint arXiv:2407.07754, 2024.
>
> Hsin-Yuan Huang, Yunchao Liu, Michael Broughton, Isaac Kim, Anurag Anshu, Zeph Landau, and Jarrod R McClean. Learning shallow quantum circuits. arXiv preprint arXiv:2401.10095, 2024.
>
> Zeph Landau and Yunchao Liu. Learning quantum states prepared by shallow circuits in polynomial time. arXiv preprint arXiv:2410.23618, 2024.
>
> Hsin-Yuan Huang, Richard Kueng, Giacomo Torlai, Victor V Albert, and John Preskill. Provably efficient
> machine learning for quantum many-body problems. Science, 377(6613):eabk3333, 2022.

---

> ### Author Response · Authors · 2024-11-20
> **Addressing "Questions/Weaknesses in Official Review of 3076 by Reviewer m7qv" (part 2)**
>
> Finally, we would like to highlight the differences between weakly noisy quantum states and pseudorandom quantum states. Specifically, we consider the second-moment statistic property of a set of pseudorandom quantum circuits $U$ whose circuit depth $\tilde{R}\leq {\rm poly}\log n$. In Ref. [T. Schuster et al., 2024], they claimed that even $\mathcal{O}(\log(n))$-depth Clifford circuits may approximate the unitary
> 2-design property given by Haar measure. Then, we may suppose $U=U_1\cdots U_{\tilde{R}}$ representing a random Clifford circuit, then for the initial state $|0^n\rangle\langle0^n|$ and arbitrary observable $O$, we have
> \begin{align}
>     M_2(U)=\int_{U\sim{\rm Cl}(2^n)}{\rm Tr}\left[U|0^n\rangle\langle0^n|U^{\dagger
>     }O\right]{\rm Tr}\left[U|0^n\rangle\langle0^n|U^{\dagger
>     }O\right]=\frac{1}{d(d^2-1)}\left[{\rm Tr}^2[O]+{\rm Tr}[O^2]\right].
> \end{align}
> On the other hand, in the context of weakly noisy environment, the Clifford circuit $U$ may transform to the channel representation $\mathcal{U}=\bigcirc_{r=1}^{\tilde R}\mathcal{E}\circ \mathcal{U}_r$ where $\mathcal{E}$ represents a $n$-qubit Pauli channel and $\mathcal{U}_r=U_r(\cdot)U_r^{\dagger}$ represents a layer of Clifford gate.
>
>  According to the ``Channel Pushing'' lemma given by Ref [Y. Quek et al., 2024] (Lemma~10), we may rewrite the noisy channel by $\mathcal{U}=\mathcal{E}^{\prime}\circ\bigcirc_{r=1}^{\tilde R}\mathcal{U}_r$,
>
> where $\mathcal{E}^{\prime}$ also represents an $n$-qubit Pauli channel. Let the channel $\mathcal{E}^{\prime}=\sum_lK_l(\cdot)K_l^{\dagger}$, where $K_l$ represents the Kraus operator satisfying $\sum_lK_l^{\dagger}K_l=I$, we have
> \begin{eqnarray}
> \begin{split}
>      M_2(\mathcal{U})=&\int_{U\sim{\rm Cl}(2^n)}{\rm Tr}\left[\mathcal{U}(|0^n\rangle\langle0^n|)O\right]{\rm Tr}\left[\mathcal{U}(|0^n\rangle\langle0^n|)O\right]
> \end{split}
> \end{eqnarray}
> \begin{align}
> =\sum\limits_{l_1,l_2}\int_{U\sim{\rm Cl}(2^n)}{\rm Tr}\left[K_{l_1}U(|0^n\rangle\langle0^n|)U^{\dagger}K_{l_1}^{\dagger}O\right]{\rm Tr}\left[K_{l_2}U(|0^n\rangle\langle0^n|)U^{\dagger}K_{l_2}^{\dagger}O\right]
> \end{align}
> \begin{align}
> =\sum\limits_{l_1,l_2}\left(\frac{1}{d(d^2-1)}\left({\rm Tr}(K_{l_1}OK_{l_1}^{\dagger}){\rm Tr}(K_{l_2}OK_{l_2}^{\dagger})+{\rm Tr}(K_{l_1}OK_{l_1}^{\dagger}K_{l_2}OK_{l_2}^{\dagger})\right)\right).
> \end{align}
> We note that the interaction term $\sum_{l_1,l_2}{\rm Tr}(K_{l_1}OK_{l_1}^{\dagger}K_{l_2}OK_{l_2}^{\dagger})$ may not equal to ${\rm Tr}(O^2)$, and this leads to $M_2(U)\neq  M_2(\mathcal{U})$. This comparison indicates that **even if an algorithm can learn the difference between noisy ensembles and Haar-random states, this does not imply that the algorithm can be used to distinguish between pseudorandom ensembles and Haar-random ensembles.**
>
>
> The above discussions demonstrate the fundamental difference between our work and Ref. [T. Schuster et al., 2024]. Such difference implies that the efficiency of our learning algorithm may not contradict the findings presented in Ref. [T. Schuster et al., 2024]. We hope these theoretical examples help clarify your initial confusion. **This discussion has been supplemented to the revised manuscript on Pages~15-16.**
>
> References:
> Yihui Quek, Daniel Stilck Fran¸ca, Sumeet Khatri, Johannes Jakob Meyer, and Jens Eisert. Exponentially
> tighter bounds on limitations of quantum error mitigation. Nature Physics, 20(10):1648–1658, 2024.

---

> ### Author Response · Authors · 2024-11-20
> **Addressing "Questions/Weaknesses in Official Review of 3076 by Reviewer m7qv" (part 3)**
>
> **Comment 2**: *Could the authors provide numerical simulations demonstrating their learning algorithm's performance on concrete examples of noisy quantum states? This would help validate the theoretical guarantees and provide intuition about the practical performance.*
>
> **Response**: We thank the reviewer for this suggestion. We have added a numerical simulation to learn from the Hamiltonian dynamics studied in Ref. [Y. Kim et al., 2023] and benchmark the complexity of noisy Hamiltonian dynamics. Due to limitations in revision time and current computational power, we may numerically test our algorithm on at most a 12-qubit density matrix (equivalent to a 24-qubit state vector), rather than on a 50-qubit density matrix, which is beyond the capabilities of the current classical simulation. We are happy to leave the large-scale algorithm simulation to future work. The numerical simulation has been supplemented to the revised manuscript. **Please refer to Sec. 6 on Page 9 and Appendix L. on Page 28.**
>
> References:
>
> Youngseok Kim, Andrew Eddins, Sajant Anand, KenXuan Wei, Ewout van den Berg, Sami Rosenblatt,
> Hasan Nayfeh, Yantao Wu, Michael Zaletel, Kristan Temme, and Abhinav Kandala. Evidence for the
> utility of quantum computing before fault tolerance. Nature, 618:500–505, 2023.

---

> ### Author Response · Authors · 2024-11-20
> **Addressing "Questions/Weaknesses in Official Review of 3076 by Reviewer m7qv" (part 4)**
>
> **Comment 3**: *The paper focuses on weakly noisy states with specific noise parameters (O(1/n) strength and O(poly log n) depth). Could the authors comment on whether similar techniques could be extended to more general noise models or stronger noise regimes? If not, could the authors describe if stronger results are simply impossible for any learning algorithm?*
>
> **Response**: We appreciate the reviewer's insightful question.
>
> In Definition 2, *we only assume the noise channel can be decomposed by a set of Kraus operators*, which is a general noise model in characterizing practical scenarios. We only require that the gate noise strength $p=\mathcal{O}(n^{-1})$ meanwhile the quantum circuit depth is constrained to be $\tilde{R}\leq {\rm poly}\log n$. These two conditions directly lead to Fact 1 and Equation 9 (on Page 5), which suggest that weakly noisy quantum states may have a constant overlap with their corresponding pure state (when $p=0$). This result enables the use of Definition 3 to characterize the complexity of weakly noisy quantum states, as $1-\epsilon$ (as defined in Definition 3) is not expected to be exponentially vanish.
>
> From this perspective, our algorithm may not be well-suited for characterizing state complexity in scenarios where the noise strength is $p=\mathcal{O}(1)$ and circuit depth $\tilde{R}\geq\Omega(\log n)$. However, this does not imply that predicting the complexity of noisy states prepared by deep quantum circuits with constant error strength is inherently hard. According to the result in Ref. [Daniel Stilck Franca et al., 2021], noisy quantum states with $p=\mathcal{O}(1)$ and $\tilde{R}\geq\Omega(\log n)$ are close to the maximally mixed state in terms of trace distance, derived by the relative entropy and Pinsker's inequality. Given this observation, for a noisy quantum state $\rho_{\tilde{R},p}$ with $p=\mathcal{O}(1)$ and $\tilde{R}\geq\Omega(\log n)$, the standard SWAP test can be used to estimate the trace distance between $\rho_{\tilde{R},p}$ and the maximally mixed state. If the trace distance is sufficiently small, then the complexity of $\rho_{\tilde{R},p}$ can be approximately characterized by that of maximally mixed state, which has a constant circuit depth complexity.
>
> Reference:
> Daniel Stilck Franca and Raul Garcia-Patron. Limitations of optimization algorithms on noisy quantum
> devices. Nature Physics, 17(11):1221–1227, 2021.

---

> > ### Author Response · Authors · 2024-11-20
> > **Concluding remarks**
> >
> > We thank the reviewer for their insightful comments and suggestions. In response, we have clarified that our work addresses different tasks compared to Ref. [T. Schuster et al., 2024], and the efficiency of our learning algorithm does not contradict T. Schuster's results. Additionally, we have supplemented our work with numerical simulations to demonstrate the effectiveness of our quantum learning algorithm. The results reveal that the complexity lower bound for weakly noisy quantum states exhibits a distinct trend compared to that of pure states. We hope these revisions and clarifications address your concerns and improve the clarity of our arguments. We trust that these changes will help in reconsidering the evaluation of our work. Thank you again for your thoughtful feedback.

---

> ### Author Response · Authors · 2024-11-22
>
> Dear Reviewer m7qv,
>
> Thanks agian for your constructive suggestions. We have made our best effort to address your comments and revised the manuscript accordingly. As the public discussion period is nearing its end, we would be delighted to hear your feedback and suggestions on our latest responses. We look forward to your reply.
>
> Best regards!

---

> ### Comment · Reviewer_m7qv · 2024-11-27
>
> I would like to thank the authors for their responses.
> Their responses have addressed my questions.

---

> > ### Author Response · Authors · 2024-11-28
> >
> > Dear Reviewer m7qv,
> >
> > We sincerely appreciate your response and are delighted that our answers have addressed your concerns.
> >
> > Best regards!

---

### Meta-Review · Area_Chair_8GVQ · 2024-12-18

**Metareview:**

Summary:
This paper introduces an efficient quantum learning algorithm for predicting the complexity of weakly noisy quantum states. The authors develop a method that leverages classical shadow representation of quantum states and provides optimal sample complexity with polynomial classical processing time, specifically designed for weakly noisy states with noise strength O(1/n) and depth O(poly log n). The work establishes important connections between learning theory and quantum state complexity in noisy environments.

Main Strengths:
The technical soundness is demonstrated through rigorous theoretical analysis, with provable guarantees on both sample and computational complexity. The approach cleverly leverages classical shadows and intrinsic properties of quantum circuit architectures to make the learning problem tractable. The work addresses an important practical challenge by studying quantum state complexity in noisy environments that better reflect real quantum devices.

Main Weaknesses:
Initially, there were concerns about the paper's claims regarding efficient complexity prediction given known hardness results in quantum complexity theory. The experimental validation was limited, lacking numerical experiments to validate theoretical claims. The initial presentation also needed improvement in explaining technical concepts and practical implementation details.

**Additional Comments On Reviewer Discussion:**

Outcomes from Author-Reviewer Discussion:
The authors provided comprehensive responses that significantly improved the paper. They clarified the fundamental difference between their work and existing hardness results, explaining how their approach focuses on specific quantum states rather than distinguishing between quantum ensembles. They added numerical simulations demonstrating the algorithm's performance on concrete examples, including tests on up to 12-qubit density matrices. The authors also enhanced technical clarity by better explaining concepts and adding practical implementation details.

Reviewer Agreement:
There was strong consensus on the technical soundness and theoretical contributions. Initially mixed views on practical applicability and experimental validation were largely resolved through the author responses and additional experiments. All reviewers acknowledged the importance of studying quantum state complexity in noisy environments.

Suggestions to Improve:
The manuscript would benefit from clearer presentation of technical concepts in the main text, more comprehensive empirical validation with larger quantum systems when possible, and extended discussion of practical implementation considerations. The authors should also clarify the relationship between different noise models and algorithm performance.

---

### Decision · Program_Chairs · 2025-01-22

Accept (Poster)